



# Lithology and orographic precipitation control river incision in the tropical Andes

Benjamin Campforts[a,b], Veerle Vanacker[c], Frédéric Herman[d], Matthias Vanmaercke[e], Wolfgang Schwanghart[f], Gustavo E. Tenorio[b,g], Patrick Willems[h], Gerard Govers[b]

[a] *Research Foundation Flanders (FWO), Egmontstraat 5, 1000 Brussels, Belgium*
   [b] *Department of Earth and Environmental Sciences, KU Leuven, Celestijnenlaan 200E, 3001 Heverlee, Belgium*
   [c] *Earth and Life Institute, Georges Lemaître Centre for Earth and Climate Research, University of Louvain, Place Louis Pasteur 3, 1348 Louvain-la-Neuve, Belgium*
   [d] *Institute of Earth Surface Dynamics, University of Lausanne, CH-1015 Lausanne, Switzerland*
[e] *Université de Liège, Département de Géographie, Clos Mercator 3, 4000 Liège, Belgium*
   [f] *Institute of Earth and Environmental Sciences, University of Potsdam, Germany*
   [g] *Facultad de Ciencias Agropecuarias, Universidad de Cuenca, Campus Yanuncay, Cuenca, Ecuador*
   [h] *Department of Civil Engineering – Hydraulics Section, KU Leuven, Kasteelpark 40 box 2448, 3001 Leuven, Belgium*

   *Correspondence to: Benjamin Campforts (benjamin.campforts@kuleuven.be)*

**Abstract.** Process-based geomorphic transport laws enable to assess the impact of rainfall variability on bedrock river incision over geological timescales. However, isolating the role of rainfall variability on erosion remains difficult in natural environments in part because the variability of rock strength and its resistance to incision are poorly constrained. Here, we explore spatial differences in the rate of bedrock river incision in the tropical Andes. The Ecuadorian Andes are characterized

by strong rainfall gradients due to orographic precipitation sourced in the Amazon basin. In addition, the tectonic configuration has generated a profound lithological heterogeneity. The relative role of either these controls in modulating river incision on millennial time scales, however, remains unclear. Using [10]Be catchment-wide erosion rates, meteorological and hydrological data, as well as data on bedrock erodibility, we provide quantitative constraints on the importance of rainfall variability and lithological variations. Explicit incorporation of rock erodibility in river incision models predicated on the stream power

equation enables us to identify a first order control of lithology on river incision rates. Rainfall variability based on a spatially and temporally explicit hydrological dataset and a stochastic-threshold river incision model explain regional differences in river incision that cannot be attributed to topographical and/or lithological variability.



## 1.     Introduction

Research on how rainfall variability and tectonic forcing interact to make a landscape evolve over time was, for a long time, limited by the lack of techniques that measure erosion rates over sufficiently long timespans (Coulthard and Van de Wiel, 2013). As a consequence, the relative role of rainfall variability and tectonic processes had to be deduced from sediment archives (e.g. Hay et al., 1988). However, whether sediment archives offer reliable proxies remains an open research question because sediment sources and transfer times to depositional sites remain largely unknown (Bernhardt et al., 2017; Romans et al., 2016). Moreover, estimates from sediment archives have been contested due to potential observation biases (Jerolmack and Paola, 2010; Sadler, 1981).

Cosmogenic radionuclides (CRN) contained in quartz minerals of river sediments provide an alternative tool for determining catchment-wide erosion rates on a routine basis (Codilean et al., 2018; Harel et al., 2016; Portenga and Bierman, 2011). In sufficiently large catchments (> 10-50 km²), detrital CRN derived erosion rates ($E_{CRN}$) integrate over timescales that average out the episodic nature of sediment supply (Kirchner et al., 2001). Hence, benchmark or natural erosion rates can be calculated for human disturbed as well as pristine environments (Reusser et al., 2015; Safran et al., 2005; Schaller et al., 2001; Vanacker et al., 2007).

Catchment-wide erosion rates have been correlated to a range of topographic metrics including basin relief, average basin gradient and elevation (Abbühl et al., 2011; Kober et al., 2007; Riebe et al., 2001; Safran et al., 2005; Schaller et al., 2001). However, in tectonically active regimes, hillslopes tend to evolve towards a critical threshold gradient which is controlled by mechanical rock properties (Anderson, 1994; Roering et al., 1999; Schmidt and Montgomery, 1995) . Once slopes approach this critical gradient, mass wasting becomes the dominant processes controlling hillslope response to changing base levels (Burbank et al., 1996). In such configuration, hillslope steepness is no longer an indication of erosion rates and topographic metrics based on hillslope relief become poor predictors of catchment wide erosion rates (Binnie et al., 2007; Korup et al., 2007; Montgomery and Brandon, 2002).

Contrary to hillslopes, rivers and river longitudinal profiles do capture changes in erosion rates (Whipple et al., 1999). Bedrock rivers in mountainous regions mediate the interplay between uplift and erosion (Whipple and Tucker, 1999; Wobus et al., 2006). They incise into bedrock and efficiently convey sediments, thus setting the base level for hillslopes and controlling the evacuation of hillslope derived sediment. Quantifying the spatial patterns of natural erosion rates in tectonically active regions requires detailed knowledge of the processes driving fluvial incision. One of the major outstanding research questions is to understand and quantify how fluvial systems respond to external rainfall variability or tectonic forcing (Armitage et al., 2018; Castelltort et al., 2012; Finnegan et al., 2008; Gasparini and Whipple, 2014; Goren, 2016; Scherler et al., 2017; Tucker and Bras, 2000).

The use of river morphological proxies, such as channel steepness ($k_{sn}$) (Wobus et al., 2006), as a predictor for catchment denudation and thus $E_{CRN}$ has successfully been applied by Safran et al. (2005) and since being applied by many others commonly identifying a monotonically increasing relationship between channel steepness ($k_{sn}$) (Wobus et al., 2006) and $E_{CRN}$ (Cyr et al., 2010; DiBiase et al., 2010; Mandal et al., 2015; Ouimet et al., 2009; Safran et al., 2005; Vanacker et al., 2015). Several authors identified a non-linear relationship between $k_{sn}$ and $E_{CRN}$ in both regional (e.g. DiBiase et al., 2010; Ouimet et al., 2009; Scherler et al., 2014; Vanacker et al., 2015) and global compilation studies (Harel et al., 2016). Theoretical models suggest that this non-linear relationship reflect the dependency of long-term river incision on hydrological and, hence, rainfall variability (Deal et al., 2018; Lague et al., 2005; Tucker and Bras, 2000). However, identifying the impact of rainfall



variability on incision rates in natural environments has only been successful for a limited number of case studies (DiBiase and Whipple, 2011; Ferrier et al., 2013; Scherler et al., 2017).

We identify two outstanding limitations hampering wide scale application of river incision models that include rainfall variability. First, hydrological data at high temporal and spatial resolutions is usually not available, but required because mountain regions are typically characterized by large temporal and spatial variation in runoff rates (e.g. Mora et al., 2014). Yet, most of the observational records on river discharge are fragmented and/or have poor geographic cover. Second, large catchments are often underlain by variable lithologies. Studies exploring the role of river hydrology in controlling river incision have mainly focused on regions underlain by rather uniform lithology (DiBiase and Whipple, 2011; Ferrier et al., 2013) or have considered lithological variations to be of minor importance (Scherler et al., 2017). However, tectonically active regions such as the Andes range, have experienced tectonic accretion, subduction, active thrusting, volcanism and denudation resulting in a highly variable litho-stratigraphic composition (Horton, 2018). Rock strength is known to control river incision rates, and is a function of its composition and lithology (Brocard and van der Beek, 2006; Lavé and Avouac, 2001; Stock and Montgomery, 1999), its rheology and fracturing due to tectonic activity (Molnar et al., 2007). If we want to use geomorphic models not only to emulate the response of landscapes to climatically regulated rainfall and/or tectonic forces but also to predict absolute erosion rates, variations in physical rock properties need to be accounted for (Attal and Lavé, 2009; Nibourel et al., 2015; Stock and Montgomery, 1999). Furthermore, these variations in rock erodibility can potentially obscure the relation between river incision and rainfall variability and more specifically the relation between long-term erosion and rainfall rates (Deal et al., 2018). Therefore, we posit that the climatic effects on erosion rates can only be correctly assessed if the geomorphic model accounts for physical rock properties and vice versa.

In this study, we assess the influence of lithological heterogeneity and rainfall variability on erosion rates in an active tectonic setting in the tropical Andes. We apply different, stream-power based models to the Paute River basin in the Ecuadorian Andes, and subsequently evaluate model performance by comparing modelled river incision rates and CRN derived erosion rates. Thereby, we aim two answer two research questions: First, do spatial variations in lithology correlate with rates of river incision? Second, are rates of river incision further modulated by rainfall variability?

## 2. River incision models

Bedrock rivers are shaped by several processes including weathering, abrasion-saltation, plucking, cavitation and debris scouring (Whipple et al., 2013). Explicitly accounting for all these processes would render models too complex for simulations over timescales relevant to understand the uplift-climate-lithology-erosion conundrum. Therefore, river incision is typically simulated by assuming a functional dependence of river incision on the shear stress ($\tau$, [Pa]) exerted by the river on its bed. Several models have been proposed to simulate the dependence of long term river incision on shear stress (Dietrich et al., 2003) where the drainage Area based Stream Power Model (A-SPM) is the most commonly used (Howard, 1994; Lague, 2014):

$$E = K'A^m S^n \qquad (1)$$

in which $E$ is the long term river erosion (L t$^{-1}$), $K'$ (L$^{1-2m}$t$^{-1}$) quantifies the erosional efficiency as a function of rock erodibility and erosivity, $A$ (L$^2$) is the upstream drainage area, $S$ [L L$^{-1}$] is the channel slope, and $m$ and $n$ are exponents whose values depend on lithology, rainfall variability and sediment load.



Eq *(1)* can be rewritten as a function of the channel steepness, $k_s$:

$$E = K' k_s{}^n \qquad (2)$$

where $k_s$ can be written as the upstream area weighted channel gradient:

$$k_s = SA^\theta \qquad (3)$$

In which $\theta = m/n$ is the channel concavity (Snyder et al., 2000; Whipple and Tucker, 1999). In order to compare steepness

indices from different locations, $\theta$ is commonly set to 0.45 and the channel steepness is referred to as the normalized steepness index, $k_{sn}$ (Wobus et al., 2006). Variations in $k_{sn}$ are often used to infer uplift patterns, by assuming a steady state between uplift and erosion (Kirby and Whipple, 2012). In transient settings, where steady state conditions are not necessarily met, the $k_{sn}$ values can be used to infer local river incision rates (Harel et al., 2016; Royden and Taylor Perron, 2013).

Notwithstanding empirical evidence supporting the A-SPM such as the scaling between drainage area and channel

slope in steady state river profiles (Lague, 2014) or its capability to simulate transient river incision pulses (Campforts and Govers, 2015), the A-SPM is a semi-empirical geomorphic 'law' with several shortcomings reviewed in Lague (2014). Most notably, the A-SPM does not explicitly simulate the effect of incision thresholds for river incision to occur (Lague, 2014), albeit numerical simulations have shown that the use of a slope exponent $n$ (Eq. (1)) greater than unity can reproduce erosion rates obtained with models explicitly accounting for incision thresholds (Gasparini and Brandon, 2011).

A state-of-the-art river incision model to simulate the impact of hydrological variability on river incision efficiency is the Stochastic-Threshold Stream Power Model (ST-SPM) (Crave and Davy, 2001; Deal et al., 2018; Lague et al., 2005; Snyder et al., 2003; Tucker and Bras, 2000). The ST-SPM explicitly acknowledges the existence of a shear stress threshold ($\tau_c$) which must be overcome to entrain sediment and bedrock. By incorporating stochasticity of the river discharge in the equation, the ST-SPM enables to simulate the frequency of erosive events and their impact on long term river incision. We

refer to literature for a full derivation of the ST-SPM (Crave and Davy, 2001; Deal et al., 2018; Lague et al., 2005; Snyder et al., 2003; Tucker and Bras, 2000).

The ST-SPM has two components. The first component involves the formulation to calculate instantaneous river incision ($I$, [L t$^{-1}$]):

$$I(Q^*) = K Q^{*\gamma} k_s^n - \psi$$

$$K = k_e k_t^a k_w^{-a\alpha} \overline{R}^m; \; \psi = k_e \tau_c^a \qquad (4)$$

$$\gamma = a\alpha(1-\omega_s); \; m = a\alpha(1-\omega_b); \; n = a\beta$$

in which $Q^*$ represents the dimensionless normalized daily discharge calculated by dividing daily discharge $Q$ [L$^3$t$^{-1}$] by

mean-annual discharge $\overline{Q}$ [L$^3$t$^{-1}$], $k_e$ [L$^{2.5}$ T$^2$ m$^{-1.5}$] is the erosional efficiency constant, $\overline{R}$ [L t$^{-1}$] is the mean annual runoff, $a$ is the shear stress exponent reflecting the nature of the incision process (Whipple et al., 2000) and $k_t$, $k_w$, $\alpha$, $\beta$, $\omega_a$ and $\omega_b$ are channel hydraulic parameters described in Table 1. The second component derives long term river erosion by multiplying the instantaneous river incision, $I$, calculated for a discharge of a given magnitude ($Q^*$) with the probability for that discharge to occur ($pdf(Q^*)$, see section 5.1.2) and subsequently integrating this product over the range of possible discharge events





specific to the studied timescale (DiBiase and Whipple, 2011; Lague et al., 2005; Scherler et al., 2017; Tucker and Bras, 2000; Tucker and Hancock, 2010):

$$E = \int_{Q_c^*}^{Q_m^*} I(Q^*)\,pdf(Q^*)dQ^* \qquad (5)$$

in which $Q_c^*$ is the minimum normalized discharge which is required to exceed the critical shear stress ($\tau_c$) and $Q_m^*$ is the maximum possible normalized discharge over the time considered.

A third river incision model further discussed in the paper, is the Runoff-based SPM (R-SPM). The R-SPM shares
its derivation with the ST-SPM but assumes river incision thresholds to be negligible ($\psi = 0$) and discharge to be constant over time ($Q^* = 1$), simplifying Eq. *(5)* to:

$$E = K k_s{}^n \qquad (6)$$

## 3.     Methods

### 3.1.     Optimization of model parameters

The presented forms of the stream power model all depend on river steepness, $k_{sn}$, known to correlate well with $E_{CRN}$
(DiBiase et al., 2010; Ouimet et al., 2009; Scherler et al., 2017; Vanacker et al., 2015). Moreover, $E_{CRN}$ integrate over timespans that average out the episodic nature of erosion and over spatial extents large enough to average out the stochastic nature of hillslope processes. Moreover, if we assume that river incision occurs at rates of catchment-wide denudation, $E_{CRN}$ can be used to constrain models of river incision (cfr. DiBiase and Whipple, 2011; Scherler et al., 2017).

To optimize model parameters, we maximize the Nash Sutcliff model efficiency (*NS*, Nash and Sutcliffe, 1970)
between observed erosion (*O*) and modelled river incision (*M*):

$$NS = 1 - \frac{\sum_{i=1}^{i=nb}(O_i - M_i)^2}{(O_i - \overline{O})^2} \qquad (7)$$

where *nb* is the number of $E_{CRN}$ samples. The *NS* coefficient ranges between $-\infty$ and 1 where 1 indicates optimal model performance explaining 100 % of the data variance. When *NS* = 0, the model is as good a predictor as the mean of the observed data. When *NS* <= 0; model performance is unacceptably low. The *NS*-coefficient has been developed in the framework of hydrological modelling but has been applied in wide range of geomorphologic studies (e.g. Jelinski et al., 2019; Nearing et
al., 2011).

### 3.2.     River incision models

In a first set of model runs, we evaluate the performance of the A-SPM in predicting $E_{CRN}$ rates. To account for rock strength variability Eq. (2) is rewritten as:

$$E = k_a\,\overline{L_E}\,k_{sn}{}^n \qquad (8)$$

where $k_a$ ($L^{1-2m}t^{-1}$) is the erosional efficiency parameter and $\overline{L_E}$ is a dimensionless catchment mean lithological erodibility
value.



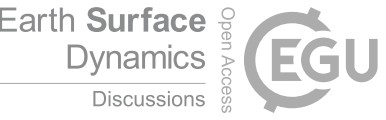

In a second set of model runs, we evaluate whether the R-SPM can explain regional differences in river incision that cannot be attributed to topographical and/or lithological variations. To account for rock strength variability Eq. (6) is rewritten as:

$$E = K\overline{L_E}k_{sn}{}^n \tag{9}$$

An overview of the parameter values required to solve the R-SPM is given in Table 1. Only the value of $k_w$ is based on a regional calibration of the hydraulic geometry scaling (see section 4.4). Other parameters are set to commonly used values (Deal et al., 2018; DiBiase and Whipple, 2011; Scherler et al., 2017). Actively incising bedrock channels are often covered by a layer of sediment. Therefore, we assume that river incision is scaled to the bed shear stress similar to bedload transport (Meyer-Peter and Müller, 1948) and set $a$ to 3/2 (cfr. DiBiase and Whipple, 2011; Scherler et al., 2017). We use the Darcy-Weisbach resistance relation and coefficients ( $\alpha = \beta = 2/3$) to calculate shear stress exerted by the river flow on its bed and assume a friction factor of 0.08 resulting in a flow resistance factor $k_t$ of 1000 kg m$^{-7/3}$ s$^{-4/3}$ (e.g. Tucker, 2004). The use of Darcy-Weisbach friction coefficients in combination with a = 3/2 results in a value for the slope exponent equal to unity ($n$ = 1, see Eq. *(4)*). Based on these theoretical derivations, we fix $n$ to unity when constraining the R-SPM. Note that this contrasts to the first set of model runs (application of the A-SPM), where we allow $n$ to vary. By fixing $n$ to unity, we want to verify whether spatial variations in runoff (incorporated in $K$ from Eq. 9) can explain variations in incision rates otherwise ascribed to non-linear river incision. The only parameter not fixed to a constant value is the erosivity coefficient $k_e$, which is optimized by maximizing the NS-coefficient (see section 3.1).

In a final set of model runs, we apply the ST-SPM (Eq. *(4)*) which is adjusted to account for rock strength variability as:

$$I = K_{st}\overline{L_E}Q^{*\gamma}k_{sn}{}^n - \psi \tag{10}$$

To derive long-term erosion rates ($E$), Eq. *(10)* is integrated over the probability density function of discharge magnitudes (Eq. (5)) which requires values for the lower ($Q_c^*$) and the upper ($Q_m^*$) limit of the integration interval. Constraining $Q_m^*$ is difficult based on observational records alone as they might miss some of the most extreme flooding events. However, when simulating incision rates over long time spans and thus considering long return times of $Q_m^*$ (>1000 y), the solution of Eq. (5) is insensitive to the choice of $Q_m^*$ (Lague et al., 2005). We therefore set $Q_m^*$ to *infinity* in all our model runs. The critical discharge ($Q_c^*$) for erosion to occur can be derived from Eq. (10) by setting $I$ equal to 0:

$$Q_c^* = \left(\frac{\psi}{K_{st}\overline{L_E}\,k_s^n}\right)^{\frac{1}{\gamma}} \tag{11}$$

The impact of spatial variations in runoff and discharge variability is evaluated by setting $\overline{R}$ and $k$ respectively to the catchments specific values or the mean of these values (listed in Table 2). Parameters left free during optimization are the erosivity coefficient $k_e$ and the critical shear stress $\tau_c^*$. Parameter values of both variables are optimized by maximizing the NS-coefficient (see section 3.1).

## 4. Study area

### 4.1. Tectonics and geomorphic setting



The Paute River is a 6530 km$^2$ transverse drainage basin: it has its source in the eastern flank of the Western Cordillera, traverses the Cuenca intramontane basin and cuts through the Eastern Cordillera before joining the Santiago river, a tributary of the Amazon (Figure 1; Hungerbühler et al., 2002; Steinmann et al., 1999). The Paute basin has a moderate relief with 90% of the slopes having hillslope gradients below 0.30 m m$^{-1}$ (Vanacker et al., 2007). Where the Paute River cuts through the Eastern Cordillera, the topography is rough with steep hillslopes (90$^{th}$ percentile of slope gradients = 0.40 m m$^{-1}$) and deeply

incised river valleys (Guns and Vanacker, 2013).

      Oblique accretion of terranes to the Ecuadorian margin during the Cenozoic, resulted in a diachronous exhumation and cooling history along the Ecuadorian Cordilleran system (Spikings et al., 2010). South of 1°30', where the Paute basin is situated, three distinct stages of elevated cooling have been reported during the Paleogene at 73-55 Ma, 50-30 Ma and 25-18 Ma, corresponding to a total cooling from ca. 300°C to ca. 60°C (Spikings et al., 2010). In the Western Cordillera, no elevated

cooling is observed during the Paleogene and extensional subsidence of the Cuenca basin allowed synsedimentary deposition of marine, lacustrine and terrestrial facies until the Middle to Late Miocene (Hungerbühler et al., 2002; Steinmann et al., 1999). The collision between the Carnegie ridge and Ecuadorian trench at some time between the Middle to Late Miocene (Spikings et al., 2001) resulted in uplift of the Western Cordillera and caused a tectonic inversion of the Cuenca basin (Hungerbühler et al., 2002; Steinmann et al., 1999). Based on a compilation of mineral cooling ages available for the Cuenca

basin, Steinman et al. (1999) estimated a mean rock uplift rate of ca. 0.7 mm yr$^{-1}$ and a corresponding surface uplift of ca. 0.3 mm yr$^{-1}$ from 9 Ma to present.

      The Paute basin is characterized by a tropical mountain climate (Muñoz et al., 2018). Despite the presence of mountain peaks up to ca. 4600 m (Figure 1), the region is free of permanent snow and ice (Celleri et al., 2007). The region's precipitation is regulated by its proximity to the pacific Ocean (ca. 60 km distance); the seasonally shifting of the Intertropical Convergence

Zone (ITCZ); and the advection of continental air masses sourced in the Amazon basin, giving rise to an orographic precipitation gradient along the eastern flank of the Eastern Cordillera (Bendix et al., 2006). Total annual precipitation is highly variable within the Paute basin and ranges from ca. 800 mm in the center of the basin, at the center of the Inter Andean valley, up to ca. 3000 mm in the eastern parts of the catchment (Celleri et al., 2007; Mora et al., 2014).

### 4.2.    CRN derived erosion rates

Catchment mean erosion rates are derived from in-situ produced $^{10}$Be concentrations in river sand. At the outlet of 30 sub-catchments indicated in Figure 1 and Table 2 (dataset published in Vanacker et al., 2015), fluvial sediments were collected. For the $^{10}$Be analysis, pure quartz was extracted from the 0.25–2.5 mm grain size fraction of the alluvial material. The $^{10}$Be was extracted from purified sand using standard methods described in von Blanckenburg et al. (1996, 2004) and the $^{10}$Be/$^9$Be ratios were measured in BeO targets with accelerator mass spectrometry at ETH Zürich. We refer to Vanacker et al.

(2015) for details on sample processing and derivation of CRN erosion rates taking into account altitude dependent production, atmospheric scaling and topographical shielding (Dunai, 2000; Norton and Vanacker, 2009; Schaller et al., 2002). CRN concentrations are not corrected for snow or ice coverage because there is no evidence of glacial activity during the integration time of CRN-derived erosion rates (Vanacker et al., 2015). Note that three data points were excluded from model optimization runs : two catchments with basin area smaller than 0.5 km² (MA1 and SA), and one catchment with an exceptionally low $^{10}$Be

concentration that can be attributed to recent landslide activity (NG-SD; see Vanacker et al., 2015).

### 4.3.    River steepness





River steepness is calculated for all channels having drainage areas of more than 0.5 km² and is averaged over 500 m reaches, based on a gap-filled SRTM v3 DEM with a 1 arc second resolution (Farr et al., 2007; NASA JPL, 2013). Because the optimized concavity $\theta$ for the Paute catchment (0.42; Text S1), is close to the frequently used value of 0.45, we fix

concavity to the reference value of 0.45 and report river steepness as normalized river steepness ($k_{sn}$) in the remainder of this paper. The spatial pattern of $k_{sn}$ values (Figure 2) is a result of the transient geomorphic response to river incision initiated at the Andes Amazon transition zone (Vanacker et al., 2015). To evaluate the extent to which transient river features influence simulated erosion rates, chi-plots ($\chi$) for all studied sub catchments are calculated following Royden and Perron, (2013) and given in the supplementary materials (Text S1; Figure S4; Royden and Taylor Perron, 2013).

**4.4.    River channel width**

Bankfull river width ($W_b$) varies with discharge as (Leopold and Maddock, 1953):

$$W_b = k_w \overline{Q}^{\omega_b} \tag{12}$$

In which $k_w$ [$L^{1-3\omega_b}t^{\omega_b}$] and $\omega_b$ are scaling parameters regulating the interaction between mean annual discharge $\overline{Q}$ and incision rates (Eq. (4)). We constrain $k_w$ by analysing downstream variations in bankfull channel width for a fraction of the river network (cfr. Scherler et al., 2017). River sections are selected based on the availability of high-resolution optical

imagery in Google Earth, and river width was derived using the ChanGeom toolset (Fisher et al., 2013a; figure S5).

The power-law fit between $Q$ and $W$ yields a value of 0.43 for the scaling exponent, $\omega_b$, with an R² of 0.51 (Figure 3). This value lies within the range of published values 0.23-0.63 (Fisher et al., 2013b; Kirby and Ouimet, 2011). To maintain a dimensionally consistent stream power model, $\omega_b$ was fixed to a value of 0.55. When doing so, the fit remains good ($R^2 = 0.5$) and we obtained a $k_w$ value of 3.7 m$^{-0.65}$s$^{0.55}$ that is used in the remainder of the paper.

**5.    Environmental drivers**

**5.1.    Rainfall**

The mean catchment runoff ($\overline{R}$) and the probability density function of daily discharge ($pdf\_Q^*$) are required to simulate long term river incision with the ST-SPM. Although measured runoff data and discharge records are available for the Paute basin (Molina et al., 2007; e.g. Mora et al., 2014; Muñoz et al., 2018), the monitoring network of about 20

meteorological and 25 hydrological stations having at least 10 years of data does not allow to capture the spatial variability present in the 6530 km² basin (Figure 1). We therefore use hydrological data derived in the framework of the Earth2Observe Water Resource Reanalysis project (WRR2; Schellekens et al., 2017) available from 1979 to 2014 at a spatial resolution of 0.25° and a daily temporal resolution (earth2observe.eu). More specifically, we use the hydrological data calculated with the global water model WaterGAP3 (Water – Global Assessment and Prognosis: Alcamo et al., 2003; Döll et al., 2003). Through

a sequence of storage equations, WaterGAP3 simulates the terrestrial part of the hydrological cycle. WaterGAP has been calibrated against data from 1319 river discharge stations monitored by the Global Runoff Data Centre (GRDC) (Schmied et al., 2014), of which 10 stations are situated in the Ecuadorian Andes. In the framework of WRR2, the WaterGAP3 is forced with ERA-Interim data and the Multi-Source Weighted-Ensemble precipitation (MSWEP) product (Beck et al., 2017).



### 5.1.1. Spatial runoff patterns

Using a global hydrological reanalysis dataset such as WaterGAP has the advantage of providing daily runoff data over several decades and makes our methodology transferable to other regions. However, a spatial resolution of 0.25° is not always sufficient to represent highly variable regional trends in water cycle dynamics over mountainous regions (Mora et al., 2014). Therefore, we downscaled the Ecuadorian WaterGAP3 data to a resolution of 2.5 km by amalgamating rain gauge data with the reanalysis product. The procedure consisted of the following steps:

(i)    The relationship between precipitation ($P$) and runoff ($R$) is constrained from the fit between monthly mean values for $P$ and $R$ available for all Ecuadorian WaterGAP 0.25° pixels (Figure 4).

(ii)    A high resolution mean annual precipitation map ($P_{RIDW}$) is calculated by downscaling the WaterGAP precipitation data ($P$) using a series of rain gauge observations (338 stations, 1990-2013) collected by the Ecuadorian national meteorological service (INAMHI; available from http://www.serviciometeorologico.gob.ec/biblioteca/). A residual

inverse distance weighting (RIDW) method is applied to amalgamate mean annual gauge data with the mean annual WaterGAP3 precipitation map. First, the differences between the gauge and WaterGAP data are interpolated using an IDW method (Figure 5). Second, the resulting residual surface is added back to the original $P$ data. A similar approach is often applied to integrate gauge data with satellite products and we refer to literature for further details on its performance (e.g. Dinku et al., 2014; Manz et al., 2016). Figure 6.a shows $P$ for the Paute region, and Figure

6.c its downscaled equivalent ($P_{RIDW}$).

(iii)    All daily precipitation data (12784 daily grids between 1979 and 2014) are downscaled to 2.5 km using the ratio between $P_{RIDW}$ and $P$, thereby assuming that the mean annual correction for precipitation also holds for daily precipitation patterns.

(iv)    The relationship between $P$ and $R$ (Figure 4) is used to derive downscaled daily runoff values from the downscaled

precipitation data for every day between 1979 and 2014.

The mean annual runoff map for the Paute basin is shown in Figure 6.b and its downscaled equivalent in Figure 6.d. Mean annual values are further used to calculate mean catchment runoff ($\overline{R}$) and the discharge variability (next paragraph) for every sub-catchment described in Table 2. The mean catchment specific runoff averaged for all catchments equals $0.82 \pm 0.35$ m yr$^{-1}$.

### 5.1.2. Frequency magnitude distribution of orographic discharges

The probability distribution of discharge magnitudes consists of two components: at low discharges, the frequency of events increases exponentially with increasing discharge (Lague et al., 2005) whereas at high discharge, the frequency of events decreases with increasing discharge following a power law distribution (Molnar et al., 2006). An inverse gamma distribution captures this hybrid behaviour and can be written as (Crave and Davy, 2001; Lague et al., 2005):

$$pdf(Q^*) = \frac{k^{k+1}}{\Gamma(k+1)} e^{-\frac{k}{Q^*}} Q^{*-(2+k)} \tag{13}$$





in which $\Gamma$ is the gamma function and $k$ is a discharge variability coefficient, $k$ represents the scale factor of the inverse gamma
distribution and ($k+1$) the shape factor. Previous studies used a single, average $k$-value to characterize regional discharge:
DiBiase and Whipple (2011) use a constant $k$ value for the San Gabriel mountains whereas Scherler et al. (2017) use a constant
$k$ value for high and low discharge but distinguish between Eastern Tibet and the Himalaya. However, given the strong
variation in temporal precipitation regimes in the Paute basin (Celleri et al., 2007; Mora et al., 2014) and the recently
recognized role of spatial hydrological variability on river incision rates (Deal et al., 2018), we explicitly evaluated the role
of temporal runoff variability by calculating catchment-specific discharge distributions from the WRR2 WaterGAP dataset.

Daily variations in discharge at the sub-catchment outlets (Figure 1) were calculated by weighing flow accumulation
with runoff ($R_{RIDW}$, see section 5.1.1). For every catchment, the complementary cumulative distribution function (*ccdf*) of the
daily discharge was fitted through the observed discharge distribution as:

$$ccdf(Q^*) = \Gamma(k/Q^*, k + 1) \qquad (14)$$

where $\Gamma$ is the lower incomplete gamma function. Figure 7 illustrates the fit between the WaterGAP derived discharge
distribution and the optimized *ccdf* for one of the catchments. Site specific discharge variability values ($k$) are calculated for
all catchments and listed in Table 2. Obtained $k$-values range between 0.8 and 1.2 with a mean of $1.01 \pm 0.12$.

### 5.1.    Geology: seismic activity and lithological strength

We aim to develop a new erodibility map for Ecuador, using an empirical, hybrid classification method. Therefore,
we combine information on the lithological composition (Aalto et al., 2006) and the age of non-igneous formations assuming
higher degrees of diagenesis and increased lithological strength for older formations (cfr. Kober et al., 2015). Adding age
information to evaluate lithological strength has advantages because lithostratigraphic units are typically composed of
different lithologies but mapped as a single entity because of their stratigraphic age. The lithological erodibility ($L_E$) is
calculated as:

$$L_E = \frac{2}{7}L'$$

$$L' = \begin{cases} \dfrac{(L_A + L_L)}{3}, non-igneous\ rocks \\[2mm] \dfrac{L_L}{2}, igneous\ rocks \end{cases} \qquad (15)$$

With $L_A$ a dimensionless erodibility index based on stratigraphic age (Figure 7), and $L_L$ a dimensionless erodibility index based
on lithological strength (Table 1), similar to the erodibility indices published by Aalto (2006). Note that $L_A$ varies between 1
(Carboniferous) to 6 (Quaternary) whereas $L_L$ ranges between 2 (e.g. granite) to 12 (e.g. unconsolidated colluvial deposits).
The lithological strength thus has a double weight, resulting in $L'$ values ranging between 1 and 6. For igneous rocks, only $L_L$
is considered assuming that the lithological strength of igneous rocks remains constant over time. For river incision parameters
to be comparable to other published ranges, $L_E$ is finally scaled around one by multiplying L' with 2/7. $L_E$ therefore ranges
between 2/7 and 14/7. A description of the lithological units, the age of the formations and their lithological strength ($L_A$, $L_l$
and $L_E$) is provided in Table S3.





Using Eq. (15) , we developed a detailed erodibility map of Ecuador (Figure S1), based on the 1M geological map of Ecuador (Egüez et al., 2017). The erodibility map was validated by comparing the $L_E$ values with field measurements (n =

9) of bedrock rheology by Basabe (1998). An overview of measured lithological strength values is provided in Table S4. Figure 9 shows good agreement ($R^2 = 0.77$) between the lithological erodibility index, $L_E$, and the measured uniaxial compressive strength confirming the validity of the classification method.

To evaluate whether seismic activity could explain differences in river incision rates, we calculated catchment mean Peak Ground Acceleration (PGA) with an exceedance probability of 10% in 50 years. PGA values are derived from a recently

published hazard assessment for South America (Petersen et al., 2018) combing assembled catalogues of earthquake frequency and size, fault geometries, seismicity rate models and ground motion models all integrated in the Global Earthquake Model (GEM; Pagani et al., 2014). PGA (g) only varies marginally within the study area (Figure 11, Table 2). Therefore, we did not consider seismic activity in the remainder of this paper although its influence should be evaluated when simulating river incision rates at larger spatial scales characterized by a stronger variability in PGA.

**6.    Results**

**6.1.    Empirical river incision model (A-SPM)**

In a first set of model runs (Table 4), we evaluate the performance of the A-SPM (Eq. (8)) to predict CRN derived erosion rates ($E_{CRN}$). When erodibility is spatially uniform, long term river incision ($E$) is a power function of the normalized river steepness $k_{sn}$, scaled by an erosion efficiency coefficient ($K'$). By optimizing the fit between E and $E_{CRN}$,

$K'$ and $n$ are constrained resulting in a NS model efficiency of 0.5 and an optimized value for $n$ of 1.06 (Figure 12.a, Table 4). When including catchment specific mean lithological erodibility values ($\overline{L_E}$), model efficiency strongly increases ($NS = 0.73$) and the optimized value of $n$ equals 1.63.

To evaluate whether including spatially varying erodibility values also increases the predictive power of the river incision model, we performed a linear Bayesian regression analysis between $E_{CRN}$ and the simulated long-term river erosion

$E$. Figure 13 shows that the posterior probability of linear regression coefficients close to one is higher and with less spread when considering spatially varying lithological erodibility values. Moreover, when $E$ is only a function of $k_{sn}$, the Bayes factor equals 1.06, in comparison to a value of ca. 1400 when $E$ is a function of both $k_{sn}$ and $\overline{L_E}$ (Table 4). This implies that a river incision model accounting for variable erodibility values is supported by the data (Jeffreys, 1998).

The importance of lithological strength in controlling the A-SPM and the $k_{sn}$-$E_{CRN}$ relation confirms that strong

metamorphic and plutonic rocks erode at significantly slower rates than lithologies which are less resistant to weathering such as sedimentary deposits of loose volcanic mixtures. The empirical rock strength classification index we developed appears to be an appropriate scaling of relative rock strength: analysis of residuals did not reveal any significant relation of residuals with lithology.

When using spatially variable, catchment specific lithological erodibility values ($\overline{L_E}$) (Figure 12.b), the $n$ coefficient of

the SPM is considerably larger than unity ($n = 1.63$) and the $k_{sn}$-$E_{CRN}$ relationship becomes non-linear, corroborating earlier findings documented in e.g. Gasparini and Brandon (2011). While this may be due to the fundamental properties of river incision and erosion processes, the shape of the relation may also be affected by spatial covariates other than lithology. In the following sections, we will investigate whether this nonlinear $k_{sn}$-$E_{CRN}$ relationship can be explained by the presence of incision thresholds, variations in runoff, or a combination of both.





## 6.2.    R-SPM and ST-SPM

The previous analysis shows that the explanatory power of the A-SPM model, and therefore the $k_{sn}$-$E_{CRN}$ relationship, strongly improves when considering spatial variations in lithological erodibility. Moreover, when considering variations in lithological erodibility, river incision is found to be non-linearly dependent on the channel slope ($S$), with $n = 1.63$. In a next

step we evaluate whether this non-linear relation can be explained by spatial and/or temporal rainfall variability and/or the existence of thresholds for river incision (Table 5).

In a first set of model runs, we evaluate the performance of the R-SPM in combination with catchment specific values for mean runoff (Table 2). When lithological variability is not considered ($\overline{L_E}$ fixed to 1, R-SPM-Scenario 1 in Table 5), the R-SPM does not perform better (NS = 0.49) than a regular A-SPM (NS = 0.50; Table 4). This illustrates that studying spatial

runoff variability is not feasible when ignoring the confounding role of lithological erodibility on erosion rates. When lithological erodibility is considered (R-SPM-Scenario 2 in Table 5), the use of the R-SPM results in a good fit between modelled river erosion and observed $E_{CRN}$ rates (Figure 14.a). Although including catchment mean runoff improves the model fit (R²=0.75), the R-SPM model overpredicts low erosion rates and underpredicts high erosion rates (Figure 14.a), resulting in a Nash Sutcliff model efficiency of 0.70 which is lower than the R².

In a second series of model runs, we evaluated the performance of the ST-SPM. Table 5 provides details on the different model set-ups. In the first three scenarios, the ST-SPM is optimized assuming a constant erodibility ($L_E$ fixed to 1). Optimized values for $\tau_c$ are close to zero in the first three scenarios, suggesting the lack of a critical incision threshold. Similar to what has been found for the R-SPM, model performance is not any better compared to the use of a simple A-SPM when not considering lithological variability. In scenario 4 and 5, catchment mean runoff ($\bar{R}$) is fixed to the average value of all

catchments (0.82 m yr⁻¹). In scenario 4, $k$ is fixed to the average value for all catchments ($k = 1.01$) whereas in scenario 5, $k$ is set to the catchment specific values as listed in Table 2. Both scenario 4 and 5 perform well with a NS value equalling 0.71. Optimized values for $\tau_c$ are ca. 30 Pa. Scenarios 4 and 5 suggest that considering the spatial variability of $k$ does not improve nor decrease the performance of the ST-SPM in the Paute basin. Given that the use of the ST-SPM with constant runoff values yields a good model fit suggests that part of the non-linear relationship between river incision and $k_{sn}$ as reported in section

6.1 can be attributed to the presence of thresholds for river incision to occur (cfr. Gasparini and Brandon, 2011). In Scenario 6 and 7, $\bar{R}$ is set to the catchment specific values derived from the WaterGAP data (Table 2). Similarly to scenario 4 and 5, using catchment specific values for $k$ does not improve model performance. Using an average $k$ value (1.01) in combination with catchment specific values for runoff results in the highest model performance of all tested scenarios (Scenario 6, NS=0.75). Optimized values for $\tau_c$ of ca. 14 -15 Pa are lower compared to scenarios 4 and 5. Figure 14.b shows the result of

Scenario 6. Contrary to the R-SPM where low erosion rates are overestimated, the ST-SPM does allow to correctly predict low erosion rates due to the consideration of an incision threshold which mainly influences simulated river erosion rates at the lower end of the spectrum.

## 7.    Discussion

### 7.1.    Are CRN derived erosion rates representative for long term river incision processes?

#### 7.1.1.    Equilibrium between river incision and hillslope denudation



Assuming an equilibrium between river incision and hillslope erosion theoretically holds for landscapes which are in a steady state or for transient landscapes characterized by rapid hillslope response (e.g. threshold hillslopes). Steady state landscapes can only be achieved under stable precipitation and tectonic settings over timescales exceeding several millions of years. Such configuration is rarely met in tectonically active regions where rivers continuously transmit new environmental

perturbations to the upper parts of the catchment (Armitage et al., 2018; Bishop et al., 2005; Campforts and Govers, 2015).

The downstream reaches of the Paute catchment are a good example of a transient landscape where a major knickzone is propagating upstream in the catchment resulting in steep threshold topography downstream of the knickzone (Figure S3 and Vanacker et al., 2015). Facing a sudden lowering of their base level, soil production and linear hillslope processes such as soil creep (Campforts et al., 2016; Vanacker et al., 2019) are not any longer able to catch up with rapidly incising rivers

(Fig. 15 in Hurst et al., 2012). In transient regions, hillslopes evolve to their mechanically limited threshold slope where any further perturbation of threshold hillslopes will result in increased sediment delivery through mass wasting processes such as rockfall or landsliding (Bennett et al., 2016; Blöthe et al., 2015; Burbank et al., 1996; Larsen et al., 2010; Schwanghart et al., 2018). Given the stochastic nature of landslides, not all threshold hillslopes will respond simultaneously to base level lowering depending on local variations in rock strength, hydrology and seismic activity (Broeckx et al., 2019). Therefore, catchments

in transient regions might experience erosion in a broad range from moderate to high rates with similar probabilities.

Thus, CRN-derived erosion rates might both overestimate and underestimate long term incision rates in these catchments. Overestimation results from the occurrence of recent, deep-seated landslide events, that deliver sediments with low CRN concentration to rivers (Tofelde et al., 2018). Underestimation might occur if long-term hillslope lowering is accomplished by landslides characterized by the occurrence of rare, large events with a return period exceeding the integration

time of CRN-derived erosion rates, (Niemi et al., 2005; Yanites and Tucker, 2010).

Longitudinal profiles of rivers draining to the knickzone in the Paute catchment show marked knickpoints (ID's 9-16 on Figure 1; Figures S3 and S4). Figure 14.b shows that simulated erosion rates for some of these catchments deviate from CRN derived erosion rates (ID's 13 14 and 16) whereas for others (e.g. ID's 9 and 11), predictions from the stochastic threshold river incision model show a good agreement with $E_{CRN}$ data. We attribute this variability to differences in drainage

area between these catchments. For catchments with a sufficiently large drainage area, modelled incision rates correspond well with $E_{CRN}$ (ID's 9 and 11 being both ca. 700 km²), most likely because the mechanisms that potentially cause overestimation and underestimation cancel each other out at this scale. For smaller catchments (ID's 8;13;14 and 16 all being < 12 km²) there is a discrepancy between simulated river incision rates and $E_{CRN}$.

Although river incision rates can be used to estimate general erosion patterns in large transient catchments (>> 10 km²),

there is a need to develop alternative approaches to simulate erosion rates in transient regions over different spatial scales. One such approach could be the explicit integration of landslide mechanisms in long term landscape evolution models such as TTLEM (Campforts et al., 2017) or Landlab (Hobley et al., 2017) to capture the stochastic nature of these processes (Niemi et al., 2005; Yanites et al., 2009).

### 420    7.1.2.    Integration timescales of $E_{CRN}$ and $k_{sn}$

CRN concentrations in detrital sediments integrate over timescales dependent on the erosion rate of the catchment. For a rock density of 2.7 g cm$^{-3}$, the integration time corresponds to the time required to erode ca. 60 cm of rock (Kirchner et al., 2001). $E_{CRN}$ in the Paute basin varies between 5 to 399 mm yr$^{-1}$ implying integration times ranging from ca. 1.5 to 175 ky. Topographical river profiles on the other hand are the outcome of the dynamic interplay between tectonics, lithology, rainfall



variability and internal drainage reorganization over timescales well exceeding one million years (Campforts et al., 2017;
Goren et al., 2014; Wobus et al., 2006).

Thus, successful identification of a rainfall variability signal is only possible if the signal has been present during the
integration timescale of both $E_{CRN}$ and $k_{sn}$. Given the high sensitivity of extreme precipitation events to climate change
(Gorman, 2012), rainfall variability over the last 10-100 ky might be well represented in $E_{CRN}$ rates but not in $k_{sn}$ values which

potentially integrate over longer timespans which are most likely characterized by important variations in hydrology.
Moreover, we use hydrological data integrating over "only" 35 years to constrain the distribution of river discharge: these
data are unlikely to fully capture rainfall variability over the integration timespan of $E_{CRN}$ measurements. Different integration
timespans of river profile response, $E_{CRN}$ rates and hydrological data can be expected to affect model performance.

While our dataset does not enable us to fully capture rainfall variability, a distinction can be made between temporal

and spatial variations. Contrary to temporal variations controlling frequency and magnitude of discharge events, the spatial
gradient in orographic precipitation is characteristic to the formation of a mountain range at geological timescales (Garcia-
Castellanos and Jiménez-Munt, 2015). In the case of the Southern Ecuadorian Andes, orographic precipitation results from
moist air advection via the South American Low-Level flow (Campetella and Vera, 2002). The air is lifted as it passes over
the eastern flanks of the Andes, resulting in moist convection fuelled by adiabatic decompression. Onset of Andean uplift in

Ecuador has been reported to be asynchronous from south to north with the onset of the most recent uplift phase dated back
to the Late Miocene (Spikings et al., 2010; Spikings and Crowhurst, 2004). Climate changes over the Miocene-Pliocene
probably altered absolute amounts of precipitation in the Ecuadorian Andes (Goddard and Carrapa, 2018) challenging the use
of present day- runoff and discharge distribution to predict long term river incision. However, the orographically induced
gradients in precipitation must have been present for timescales exceeding those represented by both $k_{sn}$ and $E_{CRN}$. This partly

explains why accounting for spatial variations in precipitation does improve the performance of a stochastic threshold SPM
contrary to the use of catchment specific discharge distributions representing temporal discharge variability.

Downscaling the WRR2 WaterGAP reanalysis dataset by amalgamating regional rain gauge data, allowed to obtain a
runoff dataset at a resolution suitable for use in our study. However, to further improve the accuracy of hydrological data, the
use of more advanced methods might be considered. A possible approach is the application of regional climate models (e.g.

Thiery et al., 2015) in regions with pronounced topographic and climatological gradients. Regional climate models have been
shown to simulate rainfall variability more realistically than global re-analysis datasets in mountainous areas (Thiery et al.,
2015) and have been successfully used to explain geomorphic response in such areas (Jacobs et al., 2016).

### 7.2. Environmental control on long term river incision rates

#### 7.2.1. Geology

Incorporating rock strength variability when simulating river incision improves model efficiency for all evaluated SPMs
(Table 4 and Table 5). Our results corroborate earlier findings that established functional dependencies between river incision
and rock physical properties to successfully determine river incision rates (Lavé and Avouac, 2001; Stock and Montgomery,
1999). In this study, rock strength is represented by an empirically derived lithological erodibility index ($L_E$, Eq. (15)) based

on the age and the lithological composition of stratigraphic units. Because of its simplicity, our empirical approach holds
potential to be applied at continental to global scales where detailed information on rock physical properties are not always
available. However, at smaller scales, studies evaluating the role of rock strength heterogeneity on specific river incision
processes such as fluvial abrasion will benefit from a more mechanistic approach to quantify rock strength (Attal and Lavé,



2009; Nibourel et al., 2015). Moreover, river incision efficacy might also depend on the density of bedrock fractures, joints
and other discontinuities (Whipple et al., 2000). Fracture density has in turn be linked to spatial patterns of seismic activity
(Molnar et al., 2007). Given the limited variability of seismic activity within the Paute basin (Petersen et al., 2018), seismicity
was not considered in our statistical regional analysis but should be considered when applying our approach to other regions
prone to more seismic variability.

We show that considering rock strength variability not only reduces the scatter surrounding the modelled river incision
versus $E_{CRN}$ derived erosion rates, but also controls the degree of the nonlinearity between river steepness ($k_{sn}$) and erosion
rates, expressed by the $n$ coefficient in the A-SPM (Figure 12). When not considering rock strength variability, the $k_{sn}$-$E_{CRN}$
relationship is close to being a linear one for the Paute catchments (with $n$ =1.06). This opposes to findings from regional
studies where lithology can be assumed uniform and $n$ has been reported to be larger than 1 (e.g. DiBiase et al., 2010; Lague,
2014; Whittaker and Boulton, 2012). In the Paute basin, the confounding role of lithology obscures a non-linear relationship
between river incision and channel steepness. Applying advanced process-based river incision models (R-SPM and ST-SPM,
Table **5**) without correction for this confounding role of lithology has proven to be of no added value in comparison to the
application of a simple, purely empirical A-SPM (Table 4 and Table *5*).

### 7.2.2.  Rainfall

After correction for lithological strength variability, a non-linear relationship between $k_{sn}$ and $E_{CRN}$ emerges (similar to
$n > 1$ in the A-SPM, Figure 12.b). With theory predicting river incision to be linearly dependent on $k_{sn}$ (Eq. (4)) when using
Darcy Weisbach friction coefficients), we evaluated whether (i) spatial variation in runoff, (ii) the existence of incision
thresholds or (iii) a combination of both can explain this nonlinearity.

Application of the R-SPM enables to include regional variations in runoff and results in a good fit and model efficiency
($R^2$=0.75, NS=0.7, R-SPM Scenario 2 in Table 5). This suggests that part of the frequently reported, non-linear relationship
between $k_{sn}$ and $E_{CRN}$ can be attributed to the spatial variability of mean annual rainfall. In tectonically active regions, steep
river reaches often appear at the edge of the mountain range where mean annual rainfall rates are high due to orographic
precipitation. Therefore, if variations in runoff are not considered, the confounding role of orographic precipitation will be
accommodated for by a non-linear relationship between river steepness and erosion rates. Application of the R-SPM does
allow to account for this effect but results in underestimation of low river incision rates (deviation from the 1:1 line on Figure
14.a). This artefact is overcome when applying the ST-SPM where the explicit simulation of a threshold improves model
performance, especially for low erosion rates.

Application of the ST-SPM assuming a constant runoff (ST-SPM Scenario 4 in Table 5), results in a slightly better
model efficiency in comparison to the R-SPM scenario with variable runoff (NS = 0.71). The latter hints at the important role
of thresholds for river incision to occur. Ultimately, the use of spatially variable runoff values in combination with the ST-
SPM, results in the best model fit and efficiency (ST-SPM Scenario 6 in Table 5, with $R^2$ =0.75 and NS = 0.75). To further
explore the interdependency between incision thresholds and spatial runoff variability, our approach can potentially be applied
to CRN datasets, covering regions characterized by more pronounced rainfall gradients (e.g. in Chile: Carretier et al., 2018).
Accounting for spatial variations in temporal discharge distributions (with $k$ characterizing the stochastic flood occurrence),
did not further improve neither deteriorate model performance (ST-SPM Scenario 7 in Table *5*). This is likely due to data
limitations: the necessary data to characterize temporal variations in discharge within a given catchment over a timescale that
is relevant for CRN-derived erosion rates are, at present, not available.



Our finding that mainly spatial patterns in precipitation control river incision patterns corroborate findings in the Himalaya (Scherler et al., 2017) and in the Andes (Sorensen and Yanites, 2019). Sorensen and Yanites (2019) evaluated the role of latitudinal rainfall variability in the Andes on erosional efficiency using a set of numerical landscape evolution model runs. They show that erosion efficiency in tropical climates at low latitudes, where the Paute basin is located, is well captured by the spatial pattern of mean annual precipitation and thus runoff. At higher latitudes (25-50°) where storms are less frequent but still very intense, mean annual precipitation decreases but erosivity is still high due to intensity of storms (Sorensen and Yanites, 2019). At these latitudes spatial variations in storm magnitude are therefore more likely to be reflected in river erosivity and thus catchment mean erosion rates than in the Ecuadorian Andes.

## 8. Conclusions and Implications for landscape evolution

An increasing number of studies and global compilations report a non-linear relationship between channel steepness and CRN derived erosion rates. Based on the growing mechanistic understanding of river incision processes, this nonlinear relationship is often attributed to the existence of incision thresholds. Rainfall variability, which is stochastic in nature, controls the frequency of river discharges large enough in magnitude to exceed these thresholds. Although the dynamic interplay between stochastic runoff and incision thresholds theoretically results in a non-linear relationship between channel steepness and erosion rates, coupling theory with field data has been proven challenging. We address this issue for a median sized basin in the Southern Ecuadorian Andes where we scrutinize the relationship between CRN derived erosion rates and river incision, simulated with three different Stream Power Models. We show that lithological variability obscures the relationship between channel steepness-based river incision and CRN derived erosion rates. When not accounting for lithological variability, a process based Stochastic Threshold SPM was not performing any better than a simple, empirical, drainage area-based stream power model. Neither could the impact of rainfall variability on river incision rates be assessed.

In order to account for the confounding role of rock strength variability, which is for the Paute basin mainly ascribed to variations in lithological strength, we propose the use of an empirical lithological strength index, based on the lithology and age of lithostratigraphic units. When considering lithological variability, the relationship between river steepness and erosion rates becomes non-linear. After integrating the empirical lithological erodibility index into the erosion efficiency coefficient of the ST-SPM, the model is capable to explain differences in subcatchment erosion rates. Considering river incision thresholds improves modelled erosion rates for slowly eroding catchments characterized by low to moderate relief. Using a downscaled version of a state-of-the-art hydrological reanalysis dataset, we furthermore show that spatial variations in runoff explain part of the variability of the observed erosion rates. The impact on river incision of temporal variations in discharge, controlling the magnitude and frequency of fluvial discharge, could not be identified within the studied catchments. We attribute this partly to the limited CRN dataset but mainly to the lack of rainfall data which integrate over sufficiently long timescales to be recorded in the CRN derived erosion rates.

Our study shows the potential of a stochastic threshold stream power model as a tool to explain regional and, potentially, continental to global differences in rainfall variability. However, the latter will only be successful after elucidating the confounding role of other environmental variables such as rock strength on river incision rates. Simplifications involved with the use of any Stream Power based incision model such as the lack of sediment-bedrock interactions or dynamic channel width adjustments might explain part of the remaining scatter surrounding predicted versus measured erosion rates. However, residual analysis showed that most of the remaining scatter occurs in small transient catchments (up to 10 km²). To further our understanding of landscape evolution over different spatial scales in such transient regions, we propose the development



of process-based landscape evolutions models explicitly simulating the coupling between transient river adjustment and stochastic hillslope response.


*Data availability.*

All data used in this paper is freely available from referenced agencies. Hydrological data is available from earth2observe.eu and http://www.serviciometeorologico.gob.ec/biblioteca/. Topographic data is available from NASA (NASA JPL, 2013). Lithological data is provided in the supplementary information. Calculations were done in MATLAB
using the TopoToolbox Software (Schwanghart and Scherler, 2014).

*Author contribution.*

In collaboration with all the authors, BC designed the project, carried out the numerical calculations and wrote the manuscript. All authors contributed to editing the manuscript.


*Competing interests.*

The authors declare that they have no conflict of interest.

*Financial support.*
B. Campforts received a postdoctoral grant from the Research Foundation Flanders (FWO)





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






**Figure 1.** Geomorphic setting of the study area. Numbered dots and corresponding watersheds indicate the sampling locations for CRN derived erosion rates (Table 2). Major faults are drawn with a full black line; *PF*: Peltetec Fault, *CF*: Cosanga Fault, *SA*: Sub-Andean thrust fault. Concealed faults separating major stratigraphical units are indicated with dashed lines. Elevations are from the 30 m SRTM v3 DEM (NASA JPL, 2013).





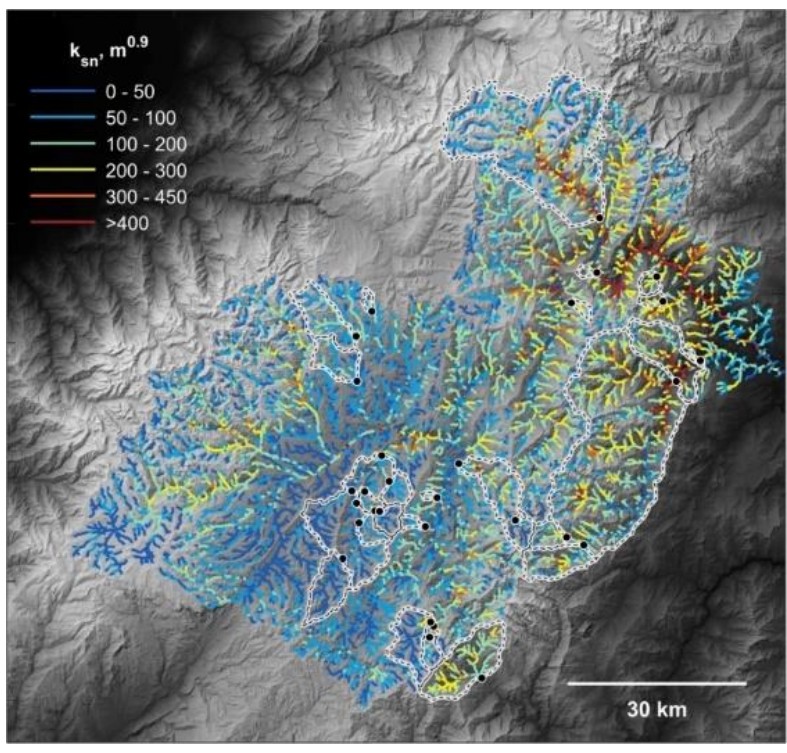


**Figure 2:** The spatial pattern of normalized steepness ($k_{sn}$) for the Paute basin overlain on hillshade map based on the 30 m SRTM v3 DEM (NASA JPL, 2013). Highest values are observed in two major knick zones in the lower part of the Paute basin where topographic rejuvenation started and a transient incision pulse has propagated from East to West, see also Figure S3.




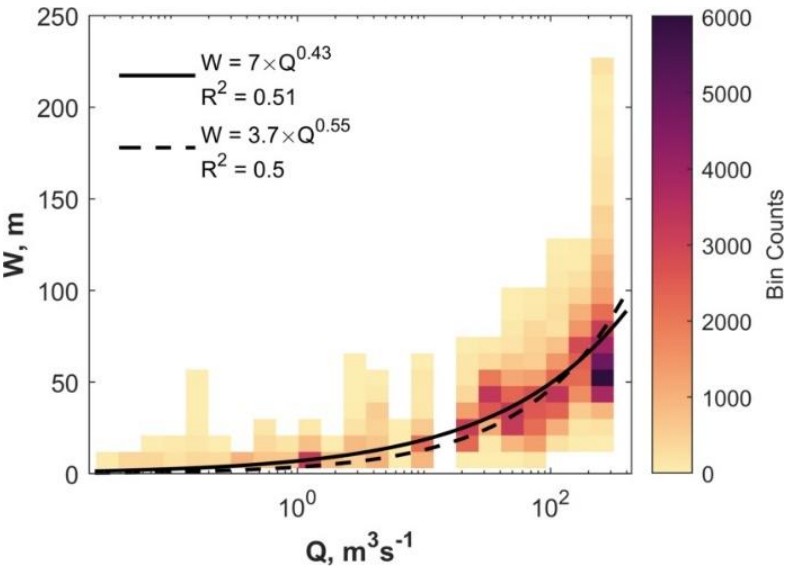

**Figure 3.** River width ($W$) as a function of the mean annual discharge ($Q$), derived from the downscaled $R_{RIDW}$ WRR2 WaterGAP3 data (available from earth2observe.eu).





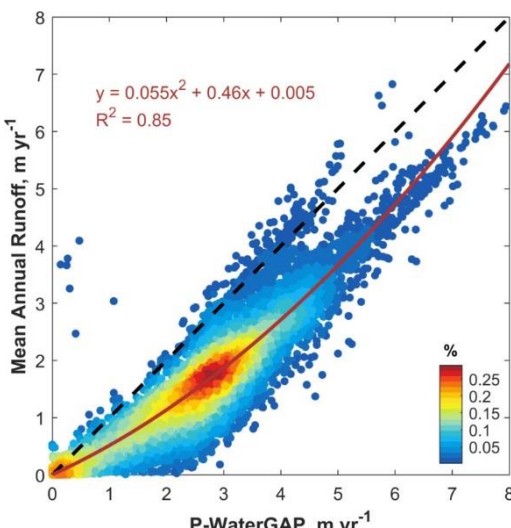


**Figure 4:** Mean monthly runoff versus mean monthly precipitation for all Ecuadorian WaterGAP3 pixels (0.25°; 1979-2014; WaterGAP3 data available from earth2observe.eu).



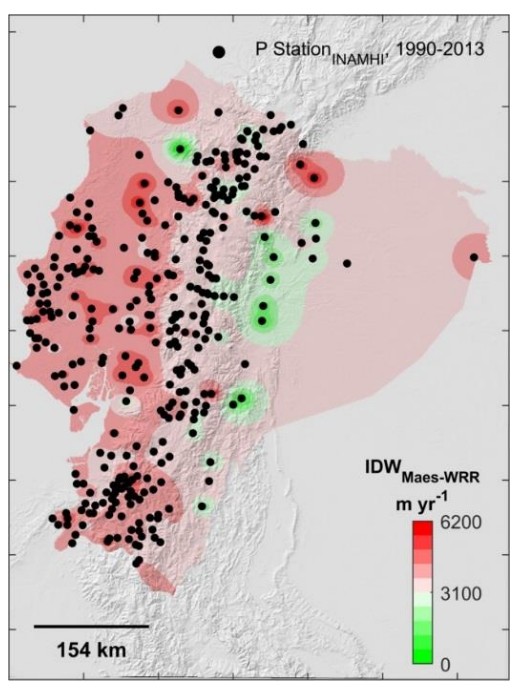


**Figure 5:** Inverse Distance Weighting (IDW) interpolation between rain gauge data (INAMHI, available from http://www.serviciometeorologico.gob.ec/biblioteca/) and WRR2-WaterGAP3 mean annual precipitation overlain on hillshade map based on the 30 m SRTM v3 DEM (NASA JPL, 2013). WaterGAP3 data available from earth2observe.eu.





**Figure 6.** Mean annual rainfall and runoff based on WRR2 WaterGAP3 data overlain on hillshade map based on the 30 m SRTM v3 DEM (NASA JPL, 2013). (a) Precipitation ($P$, 0.25°), (b) runoff ($R$, 0.25°), (c) downscaled precipitation ($P_{RIDW}$, 2500 m), (d) downscaled runoff ($R_{RIDW}$, 2500 m). WaterGAP3 data available from earth2observe.eu.






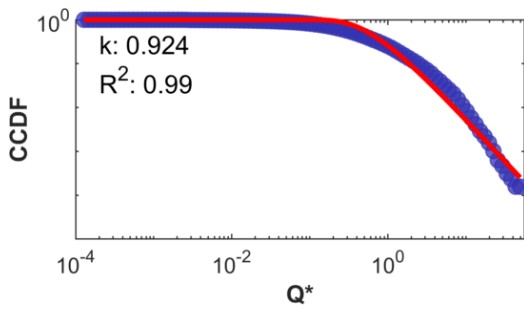

**Figure 7:** Daily discharge distribution (blue dots) derived at the outlet of one basin (NG-DW) using the downscaled WaterGAP data. The red curve depicts the fitted *ccdf* function (Eq. (14)) and its corresponding discharge variability coefficient (*k*). An overview of *k*-values for all sub-catchments is provided in Table 2.





Earth **Surface**
**Dynamics**
Discussions

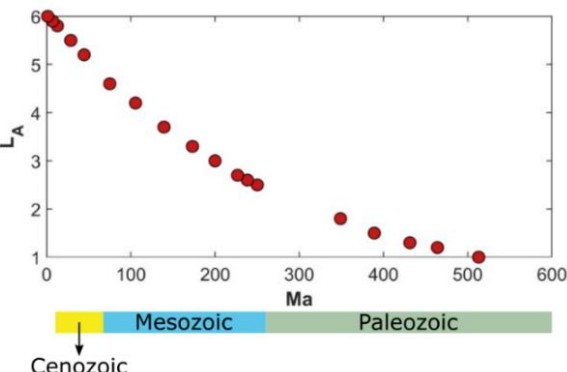

**Figure 8:** Lithological erodibility index based on lithological age ($L_A$). Detailed sub-classifications per lithology can be found in Table S1.





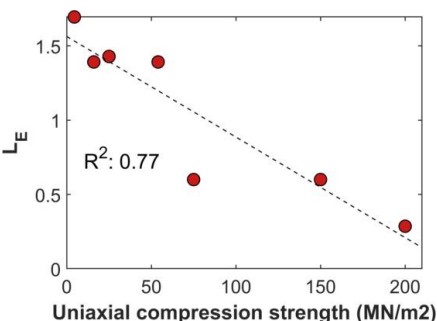

**Figure 9:** Field measurements of uniaxial compressive strength (Basabe R, 1998; Table S4) versus the empirical erodibility

index calculated using Eq. (15). Note that two out of the nine observations overlap on the plot.



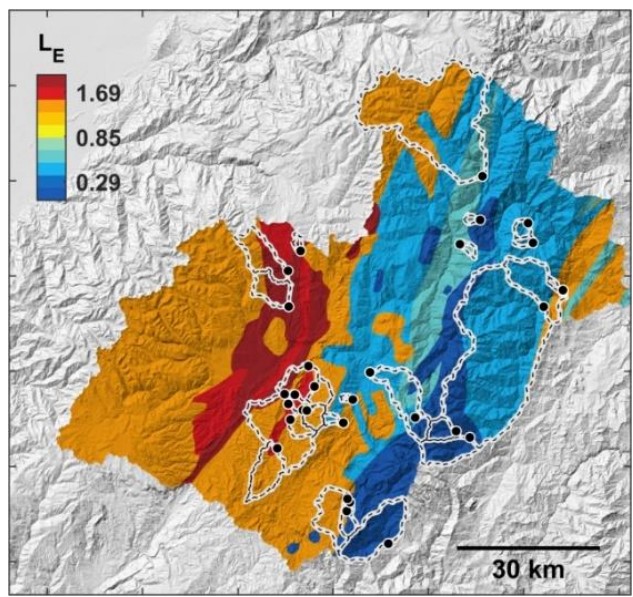

**Figure 10:** Lithological erodibility index ($L_E$) overlain on hillshade map based on the 30 m SRTM v3 DEM (NASA JPL, 2013). The map for Ecuador is shown in Figure S1.




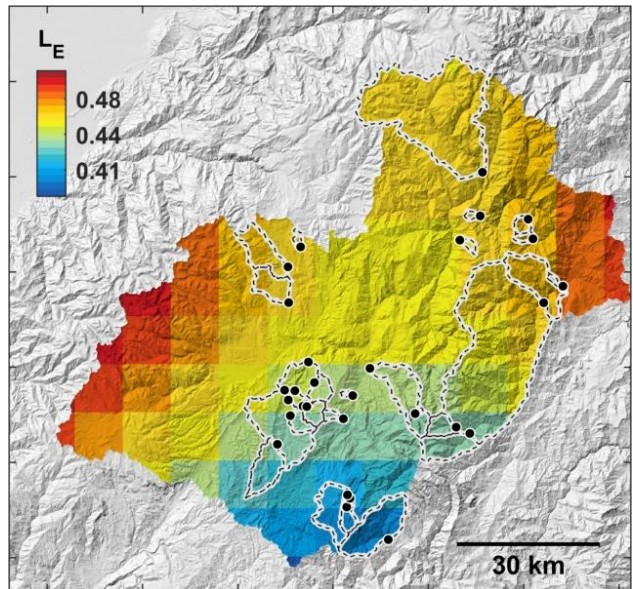

**Figure 11:** Peak Ground Acceleration (PGA, g) for a 10% probability of exceedance in a 50-year hazard level (Petersen et al., 2018) overlain on hillshade map based on the 30 m SRTM v3 DEM (NASA JPL, 2013). The map for Ecuador is shown in Figure S2.


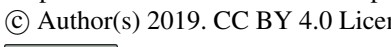



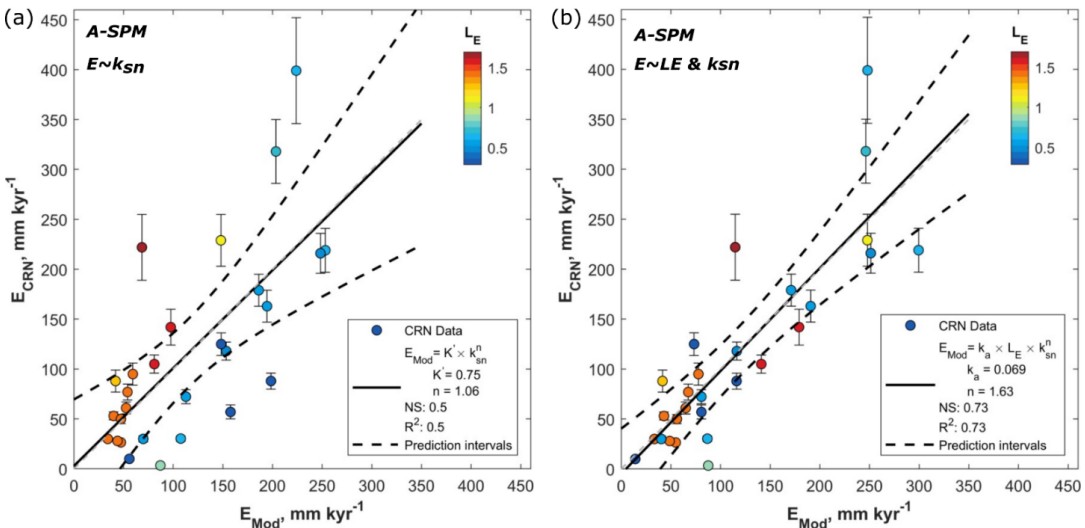

**Figure 12** Best fit between $E_{CRN}$ and modelled river incision ($E_{Mod}$) simulated using the drainage Area-based Stream Power Model (A-SPM; Eq. (8)): (a) uniform lithological erodibility ($\overline{L_E} = 1$); (b) spatially variable lithological erodibility ($\overline{L_E}$ values listed in Table 2).






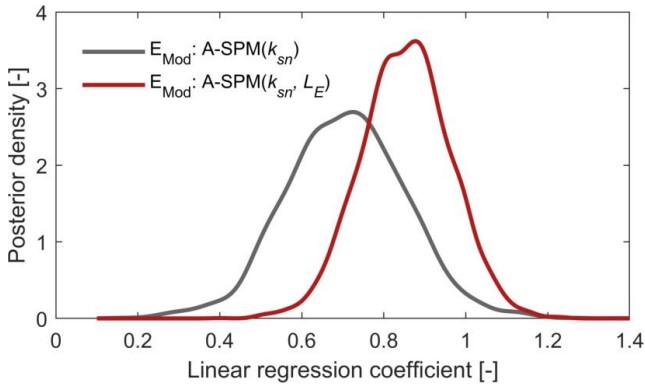

**Figure 13** Posterior probability distributions of the coefficients obtained from a linear Bayesian regression between $E_{CRN}$ and $E_{Mod}$. Bayesian regression was calculated with standardized (z-transformed) variables to enable comparison between the models.






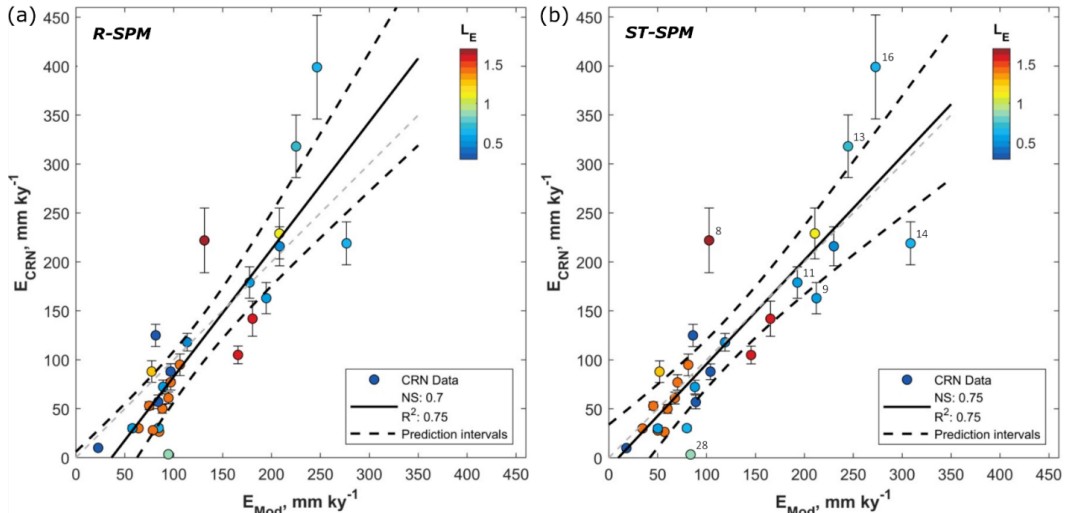

**Figure 14** Best fit between $E_{CRN}$ and modelled river incision ($E_{Mod}$) simulated using (a) the Runoff based Stream Power Model (R-SPM) and (b) the Stochastic Threshold Stream Power Model (ST-SPM). Constant model parameters are listed in Table 1 Free parameters are listed in Table 5: (a) corresponds to R-SPM Scenario 1 and (b) to ST-SPM Scenario 6. Numbered observations in (b) correspond to catchment ID's as listed in Table 2 and discussed in section 7.1.




**Table 1**: Constant parameter values used when solving the R-SPM and ST-SPM

| Parameter | Model | Description | Value | Unit |
|-----------|-------|-------------|-------|------|
| $a$ | R-SPM/ST-SPM | Bed shear stress exponent, with $\tau^a$ representing unit stream power if a= 3/2 | 3/2 | dimensionless |
| $k_t$ | R-SPM/ST-SPM | Flow resistance factor | 1000 | kg m$^{-7/3}$ s$^{-4/3}$ |
| $k_w$ | R-SPM/ST-SPM | Scaling paramter between bankfull river width and discharge | 3.7 | m$^{-0.65}$ s$^{0.55}$ |
| $\alpha$ | R-SPM/ST-SPM | Flow resistaence exponent (Darcy–Weisbach) | 2/3 | dimensionless |
| $\beta$ | R-SPM/ST-SPM | Flow resistaence exponent (Darcy–Weisbach) | 2/3 | dimensionless |
| $\theta_{\mathrm{ref}}$ | R-SPM/ST-SPM | Reference concavity | 0.45 | dimensionless |
| $\rho_s$ | ST-SPM | Sediment particle density | 2.7 | g cm$^{-3}$ |
| $\rho_w$ | ST-SPM | Fluid density | 1 | g cm$^{-3}$ |
| $\tau_c{}^*$ | ST-SPM | Shield's number | 0.045 | dimensionless |
| $\omega_b$ | ST-SPM | downstream channel width variation exponent | 0.55 | dimensionless |
| $\omega_s$ | ST-SPM | At-a-station channel width variation exponent | 0.25 | dimensionless |





**Table 2**: Properties of the sub-catchments studied in this paper. ID's correspond to the numbers indicated on Figure 1. The
$^{10}$Be cosmogenic nuclide derived erosion rates are derived from Vanacker et al. (2015)[a]. Coordinates are given in decimal
degrees in the WGS84 datum, $\overline{L_E}$ is the catchment average lithological index, $\overline{PGA}$ is the catchment average seismicity, $k_{sn}$ is
the normalized catchment average steepness, $P_{RIDW}$ and $R_{RIDW}$ are respectively the catchment average downscaled precipitation
and runoff and $k$ is the optimized discharge variability coefficient.

| ID | Sample | Lat, ° | Lon, ° | Area, km² | $^{10}$Be erosion, mm ka$^{-1}$ | $\overline{L_E}$ | $\overline{PGA}$, g | $k_{sn}$, m$^{0.9}$ | $P_{RIDW}$, m yr$^{-1}$ | $R_{RIDW}$, m yr$^{-1}$ | $k$ |
|---|---|---|---|---|---|---|---|---|---|---|---|
| 1 | BQ | -2.94 | -78.93 | 186.3 | 53 ± 4 | 1.44 | 0.44 | 41.78 | 1.06 | 0.55 | 1.18 |
| 2 | CH | -3.22 | -78.74 | 86 | 88 ± 8 | 0.34 | 0.42 | 187.79 | 1.59 | 0.87 | 0.87 |
| 3 | CJ | -2.92 | -78.88 | 19.5 | 95 ± 11 | 1.43 | 0.44 | 60.45 | 1.02 | 0.54 | 1.04 |
| 4 | DE2 | -2.77 | -78.93 | 39.1 | 105 ± 9 | 1.61 | 0.45 | 80.96 | 1.14 | 0.58 | 1.04 |
| 5 | JA21 | -2.89 | -78.89 | 276 | 50 ± 4.5 | 1.45 | 0.44 | 48.96 | 1.05 | 0.55 | 1.19 |
| 6 | MAR | -3.04 | -78.95 | 49.8 | 30 ± 2 | 1.43 | 0.43 | 35.97 | 1.07 | 0.56 | 1.08 |
| 7 | NA1 | -2.70 | -78.92 | 57.1 | 142 ± 18 | 1.54 | 0.45 | 96.36 | 1.04 | 0.53 | 1.05 |
| 8 | NA4 | -2.67 | -78.90 | 4.9 | 222 ± 33 | 1.69 | 0.45 | 69.19 | 0.87 | 0.44 | 1.11 |
| 9 | NG-DW | -2.73 | -78.40 | 686.8 | 163 ± 16 | 0.57 | 0.45 | 184.21 | 2.25 | 1.33 | 0.92 |
| 10 | NG-SD | -2.73 | -78.39 | 3.3 | 3959 ± 3801 | 0.89 | 0.46 | 231.84 | 2.62 | 1.60 | 0.91 |
| 11 | NG-UP | -2.78 | -78.46 | 679.1 | 179 ± 16 | 0.55 | 0.44 | 176.77 | 2.21 | 1.31 | 0.91 |
| 12 | PA | -2.52 | -78.56 | 424.4 | 229 ± 26 | 1.13 | 0.45 | 142.61 | 1.14 | 0.60 | 1.16 |
| 13 | PAL | -2.65 | -78.61 | 6.2 | 318 ± 32 | 0.69 | 0.45 | 192.24 | 1.89 | 1.11 | 0.88 |
| 14 | PT-BM | -2.65 | -78.46 | 6.8 | 219 ± 22 | 0.60 | 0.45 | 236.09 | 2.50 | 1.51 | 0.91 |
| 15 | PT-QP | -2.61 | -78.57 | 3.4 | 216 ± 20 | 0.52 | 0.45 | 231.77 | 2.01 | 1.16 | 0.94 |
| 16 | PT-SD | -2.61 | -78.46 | 11.1 | 399 ± 53 | 0.60 | 0.45 | 210.28 | 2.52 | 1.51 | 0.93 |
| 17 | QU | -2.99 | -78.92 | 16.7 | 77 ± 8 | 1.43 | 0.44 | 55.32 | 1.02 | 0.53 | 1.17 |
| 19 | RG1_2 | -2.96 | -78.89 | 0.9 | 26.5 ± 2 | 1.43 | 0.44 | 48.87 | 1.01 | 0.53 | 1.13 |
| 20 | RG2 | -2.94 | -78.91 | 29.2 | 61 ± 6 | 1.44 | 0.44 | 53.96 | 1.01 | 0.53 | 1.12 |
| 21 | RGD1 | -2.94 | -78.80 | 2.2 | 30 ± 3 | 0.64 | 0.44 | 105.63 | 1.03 | 0.55 | 1.14 |
| 18 | RGST | -2.97 | -78.90 | 20.2 | 28 ± 2 | 1.42 | 0.44 | 45.55 | 1.00 | 0.52 | 1.08 |
| 22 | SA | -2.96 | -78.93 | 0.5 | 152 ± 19 | 1.49 | 0.44 | 0.04 | 1.05 | 0.55 | 1.16 |
| 23 | SF1_2 | -2.89 | -78.77 | 84 | 72 ± 7 | 0.56 | 0.44 | 110.46 | 1.42 | 0.78 | 0.83 |
| 24 | SF2 | -2.98 | -78.69 | 1.3 | 118 ± 9 | 0.50 | 0.44 | 147.45 | 1.60 | 0.89 | 0.80 |
| 25 | SI1 | -3.16 | -78.81 | 0.6 | 10 ± 1 | 0.29 | 0.42 | 57.09 | 1.34 | 0.72 | 0.95 |
| 26 | SI2 | -3.14 | -78.81 | 18.3 | 30 ± 3 | 0.58 | 0.42 | 70.42 | 1.38 | 0.74 | 0.99 |
| 27 | SI3 | -3.14 | -78.81 | 49.2 | 88 ± 11 | 1.30 | 0.42 | 43.63 | 1.28 | 0.68 | 1.03 |
| 28 | SI5 | -3.00 | -78.81 | 6 | 3.4 ± 0.3 | 0.90 | 0.43 | 86.62 | 0.99 | 0.53 | 1.09 |
| 29 | TI11 | -3.01 | -78.57 | 62.1 | 125 ± 11 | 0.33 | 0.43 | 142.87 | 1.97 | 1.13 | 0.84 |
| 30 | TI2 | -3.01 | -78.61 | 21 | 57 ± 7 | 0.33 | 0.43 | 151.34 | 1.86 | 1.06 | 0.83 |

[a] Catchment MA1 from Vanacker et al. 2015 is not listed because its area (< 0.1km²) does not allow to accurately calculate
basin properties listed here

[b] Catchments excluded from model optimization runs (see text)





**Table 3**: Lithological erodibility index based on lithological strength ($L_L$). Detailed sub-classifications per lithology can be
found in Table S2.

| | $L_L$ |
|---|---|
| Igneous | 2 - 3 |
| Metamorphic (Igneous) | 2 |
| Metasedimentary | 2 - 4 |
| Strong sedimentary | 4 |
| Weak sedimentary | 10 - 12 |
| Unconsolidated | 12 |



**Table 4**: Best Fit Model Results: A-SPM

| Model | Scenario | Figure | Erosional efficiency | Erosional efficiency | Slope exponent | Bayes factor | R² | Nash Suttcliff |
|---|---|---|---|---|---|---|---|---|
| | | | $K'$ | $k_a$ | $n$ | | | NS |
| | | | $m^{0.1}s^{-1}$ | $m^{0.1}s^{-1}$ | | | | |
| A-SPM | Constant rock erodibility ($\overline{L_E} = 1$) | 12.a | 0.75 | - | 1.06 | 1.06 | 0.50 | 0.50 |
| | Variable rock erodibility | 12.b | - | 0.069 | 1.63 | 1457 | 0.73 | 0.73 |






**Table 5** Best Fit Model Results: R-SPM and ST-SPM

| Model | Scenario nb. | Description | Figure | Erosional efficiency | Discharge variability | Critical Shear stress | Runoff | R² | Nash Suttcliff |
|---|---|---|---|---|---|---|---|---|---|
| | | | | $ke$ | $k$ | $\tau_c$ | $R$ | | NS |
| | | | | $m^{2.5} s^2 kg^{-1.5}$ | | Pa | m yr-1 | | |
| R-SPM | 1 | $\overline{L_E}$ fixed[a] | - | $8.86 \times 10^{-15}$ | - | - | | 0.51 | 0.49 |
| | 2 | $\overline{L_E}$ variable[a] | 14.a | $1.43 \times 10^{-14}$ | - | - | | 0.75 | 0.70 |
| ST-SPM | 1 | $\overline{L_E}$ fixed[a] $\overline{R}$ fixed[b] $k$ fixed[c] | - | $1.14 \times 10^{-14}$ | 1.01 | 4.89 | 0.82 | 0.50 | 0.50 |
| | 2 | $\overline{L_E}$ fixed[a] $\overline{R}$ variable[b] $k$ fixed[c] | - | $9.76 \times 10^{-15}$ | 1.01 | 0.00 | *variable* | 0.51 | 0.49 |
| | 3 | $\overline{L_E}$ fixed[a] $\overline{R}$ variable[b] $k$ variable[c] | - | $9.88 \times 10^{-15}$ | *variable* | 0.00 | *variable* | 0.52 | 0.50 |
| | 4 | $\overline{L_E}$ variable[a] $\overline{R}$ fixed[b] $k$ fixed[c] | - | $2.86 \times 10^{-14}$ | 1.01 | 30.74 | 0.82 | 0.72 | 0.71 |
| | 5 | $\overline{L_E}$ variable[a] $\overline{R}$ fixed[b] $k$ variable[c] | - | $2.90 \times 10^{-14}$ | *variable* | 30.87 | 0.82 | 0.71 | 0.71 |
| | 6 | $\overline{L_E}$ variable[a] $\overline{R}$ variable[b] $k$ fixed[c] | 14.b | $1.86 \times 10^{-14}$ | 1.01 | 14.21 | *variable* | 0.75 | 0.75 |
| | 7 | $\overline{L_E}$ variable[a] $\overline{R}$ variable[b] $k$ variable[c] | - | $1.88 \times 10^{-14}$ | *variable* | 14.66 | *variable* | 0.75 | 0.75 |

[a] If $\overline{L_E}$ is fixed, a uniform value of 1 is used for all catchments. If $\overline{L_E}$ is variable, catchment specific values for $L_E$ are used (Table 2)

[b] If $R$ is fixed, a uniform mean runoff value of 0.8 m yr⁻¹ is used for all catchments. If $R$ is variable, catchment specific values are used (Table 2)


[c] If $k$ is fixed, a uniform mean discharge variability value of 1.01 is used for all catchments. If $k$ is variable, catchment specific values are used (Table 2)