# Peer review of "Parameterization of river incision models requires accounting for environmental heterogeneity: insights from the tropical Andes"

_Earth Surface Dynamics, 2019_

## Referee Comment (RC1) · Anonymous Referee #1 · 17 Nov 2019

General comments

Dear Authors, Overall I found the manuscript scientifically interesting, well written and structured. The topic is of interested for the geomorphological community, however its acceptance could be strengthened after minor corrections (see details below).

Scientific comments 1. I would suggest that the authors use a different misfit function for calculating the fit of the model to the data (see details in technical corrections). 2. It is not clear if the gained conclusions are applicable or transferable to other settings and therefore how much impact the manuscript will have in the community. The scientific relevance could be significantly strengthened if other available datasets are compared to the presented study (e.g. from DiBiase or Carritier in the the San Gabriel Mountains and the Andes). I hope you find my comments and suggestions helpful.

[Figure]

Technical corrections: Line 16-27: Since there is not word limit on the Abstract you should give some more details here. For instance, what are the erosion rates and how they differ in different lithologies/rainfall? Would be nice to have some absolute or relative values on erosion/incision depending on lithology/rainfall. Line 38: I would not give a fixed minimum catchment area since this is site-to-site depending, e.g. Kober et al. (2012) or West et al. (2014) found that nuclide concentrations of larger catchments are perturbed by single mass-wasting events. Line 42: Change to '. . . have been found to correlate with a . . .'. Line 55: Delete 'external'. Line 58-62: Please rewrite/reorder this sentence. Line 144: I would suggest to use a different misfit function, since the result is depending on the distribution of measured erosion rates and does not take into account the analytical uncertainties. Use a simple misfit function such as: Misfit=$\sum_{(I=1...nb)}\sqrt{(((O\_i-M\_i)/E\_i)\hat{}2)}$ A misfit of nb or smaller would indicate that you predict the observations within the e.g. 1 standard deviations of all observations (if E is the standard devation) and a value of 2*nb would mean you are within 2 standard deviations . . . Equation (10): Not sure, but have you explained what Kst is? Equation (11): I guess it should be ksn and not ks. Line 182: Please refer to the corresponding equations (4). Line 184: Please make sure that all local names of locations, mountain ranges, basins. . .. are shown in a figure for those reader that are not familiar with the geological/geographic setting. Line 216: A recent paper (DiBiase et al. 2018) showed that TCN do not need to be corrected for topographic shielding because of deep non-vertical attenuation paths. Line 378: Would be nice to show that the fits to your data are statistically different for your different complex models. Visually they are look very similar and if I take the confidence intervals shown that overlap. Line 384: I would not use a chapter heading without text. Line 391: In addition to the supplementary figure please add the position of knickpoints in one of your maps. Line 393: Is the baselevel lowering or the uplift increasing, please clarify! Line 430: Why do you assume that hydrological/climate changes occurred more likely on Myr-timescale compared to timescales erosion rates are averaging over? Please explain this. Line 432: Add '. . .timespan of ECRN and ksn measurements.'

ESurfD

Interactive
comment
Table 1: Change to 'Flow resistance...' Figure 1: The faults and labelling of faults is difficult to see. Larger line width and fonts, maybe even colour would help. Please show the main streams as lines. Figure 5: Add coordinates.

---

## Referee Comment (RC2) · Anonymous Referee #2 · 16 Jan 2020

General Comments: Given the focus on rainfall variability in the introduction text, I expected a paper that would advance our knowledge on the impact of rainfall variability on long term incision rates. Essentially what I read was a paper that concludes that lithological strength variability is very important in correctly predicting erosion rates and that accounting for rainfall variability also helps some (results in table 5, especially). I think the introduction needs to be revised somewhat to better reflect the results presented in the paper. The abstract does a better job of communicating the essence of the paper. Generally, the manuscript is very heavy on the methodology and too light on the discussion of the results and why these results matter. I also think the authors sometimes overreach on the significance of some results. It seemed like a long slog through the methodology section with many figures that did not seem terribly

relevant OR were uninterpretable (Figures 3,5,7,8,9,11,13). Not all of these need to be relegated to Supplementary Material, but it would be helpful if some of them were and the important figures referenced more prominently in the text. I often felt like I had to hunt down the authors motivation for a methodology or intuit the reasons why results were significant. The authors need to be clearer through out the manuscript on both of these points. With some substantial improvements to this manuscript, particularly in cutting down the methodology section and refining and expanding the results section, I think it can be published as a valuable contribution to the geomorphology community. I have many specific comments on science issues and several technical corrections that are included in an annotated PDF that I will attach.

Please also note the supplement to this comment:
https://www.earth-surf-dynam-discuss.net/esurf-2019-48/esurf-2019-48-RC2-supplement.pdf

**Supplement:**

[revised manuscript text omitted]

---

## Author Response (AR1)

**Reviewer #1**

We thank the reviewer for the thoughtful and constructive comments. In the following we address the comments and suggestions.

**General comments**

Overall, I found the manuscript scientifically interesting, well written and structured. The topic is of interested for the geomorphological community, however its acceptance could be strengthened after minor corrections (see details below).

We are pleased that the reviewer appreciates our work and sees it as a contribution to the community. In the following, we address his specific comments.

1. I would suggest that the authors use a different misfit function for calculating the fit of the model to the data (see details in technical corrections).

We think it is an interesting suggestion to calculate a model performance metric which considers the analytical uncertainty on the observed data ( $E_{CRN}$ ). However, errors on CRN data are heteroscedastic: they systematically increase with increasing erosion rates. Although the *ME* thus provides a good metric to evaluate overall model performance, the metric is not well suited to optimize model parameters in an optimization procedure: during the optimization of the model, too much weight will be given on the lower regime of the erosion spectrum where the analytical errors on  $E_{CRN}$  are low whereas the higher  $E_{CRN}$  data will not be approximated well because of their large associated errors. To compensate for the effect of heteroscedasticity we rescale values of  $O_i$ ,  $M_i$  and  $E_i$  using a logarithm with base 10 when calculating *ME*. In the revised version of the paper, the *ME* will be reported as a metric to evaluate model performance, but not to optimize model parameters. Model optimization is done using the Nash Sutcliff model efficiency, and we will explain this in the revised version of the manuscript.

2. It is not clear if the gained conclusions are applicable or transferable to other settings and therefore how much impact the manuscript will have in the community. The scientific relevance could be significantly strengthened if other available datasets are compared to the presented study (e.g. from DiBiase or Carritier in the the San Gabriel Mountains and the Andes). I hope you find my comments and suggestions helpful.

We propose a methodology for studying the spatial variability of river incision rates which can be used as a framework to study the coupling between river incision, lithological heterogeneity and climate at larger continental to global scales. However, developing a regional erodibility index and compiling hydrological datasets for regions others than the one studied here would be a project on its own and is therefore beyond the scope of this paper. In the revised version of the paper, we do stress that our findings are based on a study case and that the significance of our results should be tested by applying a similar methodology to continental or global scales.

**Technical corrections**

Line 16-27: Since there is not word limit on the Abstract youshould give some more details here. For instance, what are the erosion rates and howthey differ in different lithologies/rainfall? Would be nice to have some absolute or rel-ative values on erosion/incision depending on lithology/rainfall.

We will follow the suggestion of the reviewer to extend the abstract. However, since reviewer 2 requested more clarification on the main objectives and conclusions of our paper, we will elaborate the abstract along those lines rather than giving specific values.

Line 38: I would notgive a fixed minimum catchment area since this is site-to-site depending, e.g. Koberet al. (2012) or West et al. (2014) found that nuclide concentrations of larger catch-ments are perturbed by single mass-wasting events.

We will remove the minimum catchment area as suggested

*Line 42: Change to '. . . have been found to correlate with a. . .'.* Noted, we will revise.

*Line 55: Delete 'external'.* Noted, we will revise.

*Line 58-62: Pleaserewrite/reorder this sentence.* Noted, we will revise.

Line 144: I would suggest to use a different misfit func-tion, since the result is depending on the distribution of measured erosion rates and does not take into account the analytical uncertainties. Use a simple misfit function such as:  $Misfit=\sum_{i=1}^{I} (I=1...nb) \vee (((O_i-M_i)/E_i)^2) A$  misfit of nb or smaller would indicate that you predict the observations within the e.g. 1 standard deviations of all observations (if E is the standard deviation) and a value of 2\*nb would mean you are within 2 standard deviations...

$$\sum_{I=1}^{nb} \sqrt{\frac{\left(0\_i - M\_i\right)^2}{E\_i}}$$

See reply general comment 1.

Equation (10): Not sure, but have you explained whatKst is? Thanks for pointing this out, should be K. We will revise.

Equation (11): I guess it should be ksn and not ks. Thanks for pointing this out, should be  $k_{sn}$ . We will revise.

*Line 182: Please refer to the corresponding equations (4).* Noted, we will revise.

Line 184: Please make sure that all local names of locations, mountain ranges, basins. . .. are shown in a figure for those reader that are not familiar with the geological/geographic setting. Noted, we will adjust Figure 1.

Line 216: A recent paper (DiBiaseet al. 2018) showed that TCN do not need to be corrected for topographic shielding because of deep non-vertical attenuation paths. Thanks for pointing us to this paper. Since our paper uses the data as processed in Vanacker et al. 2015 (where a correction was applied), we will keep this section as it is.

Line 378: Would be nice to show that the fits to your data are statistically different for your different complex models. Visually they are look very similar and if I take the confidence intervals shown that overlap.

We agree: the fits for the different scenarios are similar. We feel that our sample size does not warrant a thorough statistical analysis. However, we will add the following sentences to the revised version of the paper:

"Note that differences in model performance between R-SPM scenario 2 and ST-SPM scenarios 5-8 are existent but not very pronounced. To evaluate the significance of these differences, our analysis should be repeated on larger datasets capturing a wider variability in erosion rates and hydrology"

*Line 384: I would not use a chapter heading without text.*

Given the different topics covered in the discussion section, we feel the use of subsections is warranted here to structure the flow of the paper and to keep the overview.

*Line 391: In addition to the supplementary figure please add the position of knickpoints in one of your maps.*

Good suggestions, we will adjust the figure.

*Line393: Is the baselevel lowering or the uplift increasing, please clarify!* Here we refer to the effect of propagating pulses of river incision. We will clarify: "Facing a sudden lowering of their base level after river rejuvenation, ..."

Line 430: Why do you assume that hydrological/climate changes occurred more likely on Myrtime scale compared to timescales erosion rates are averaging over? Please explain this. We do not know for sure, but given that  $k_{sn}$  values integrate over several thousands to millions of years, and CRN data only over 100-100k years, it is *more* likely that the climate has changed over the integration time captured in river steepness than over the time represented by CRN data. We will clarify as such in the text.

*Line432: Add '. . .timespan of ECRN and ksn measurements.'* Noted, we will revise.

Table 1: Change to 'Flow resistance. . .'

Noted, we will revise.

Figure 1: The faults and labelling of faults is difficult to see. Larger line width and fonts, maybe even colour would help. Please show the main streams as lines.

Good suggestions, we will adjust the figure.

Figure 5: Add coordinates.

Figure will be moved to the SI in the new version of the paper.

**Reviewer #2**

**We thank the reviewer for the thoughtful and constructive comments.**

**General comments**

Given the focus on rainfall variability in the introduction text, I expected a paper that would advance our knowledge on the impact of rainfall variability on long term incision rates. Essentially what I read was a paper that concludes that lithological strength variability is very important in correctly predicting erosion rates and that accounting for rainfall variability also helps some (results in table 5, especially).

The reviewer her/his main concern is on the role of lithology versus rainfall variability in controlling erosion rates. The reviewer concludes that lithology dominantly controls erosion and that rainfall helps some in explaining spatial patterns of erosion. In fact, that is indeed one way of looking at the problem: The Area-Based Stream Power Model (A-SPM) does a good job in predicting spatial pattern of erosion rates after correcting for lithological heterogeneity. However, the goal of this paper is not only to come up with just *a* model that describes the spatial variation in erosion rates. What we aim to do, is to explain the spatial pattern of erosion rates and to identify the factors controlling it. Therefore, we do not propose to use the R-SPM and ST-SPM erosion models as tools to 'better' predict incision, rather we use them as tools to get additional insights in the existence of a non-linear relationship between CRN-derived erosion rates and river steepness ( $k_{sn}$ ). That said, we agree with the reviewer that we were not entirely clear in passing on that message to the reader, which is why we will rewrite those parts of the manuscript where we justify the use of the different incision models. In the abstract, for example, we will add the following lines to clarify this:

" ... First, we use an area-based stream power model to scrutinize the role of lithological heterogeneity on river incision rates. We show that lithological heterogeneity is key to predicting spatial patterns of incision rates. Accounting for lithological heterogeneity reveals a non-linear relationship between river steepness, a proxy for river incision, and cosmogenic radio nuclide (CRN) derived denudation rates. Second, we explore this nonlinearity using runoff-based and stochastic-threshold stream power models, combined with a state-of-the-art hydrological dataset to calculate spatial and temporal runoff variability. Statistical modelling suggests that the non-linear relationship between river steepness and denudation rates can be attributed to a spatial runoff gradient and incision thresholds. Our findings have two main implications for the overall interpretation of CRN-derived denudation rates and the use of river incision models : (i) applying sophisticated stream power models to explain denudation rates at the landscape scale is only relevant when accounting for the confounding role of environmental factors such as lithology and (ii) spatial patterns in runoff due to orographic precipitation in combination with incision thresholds explain part of the non-linearity between river steepness and CRN-derived denudation

rates. The methodology that we present can be used as a framework to study the coupling between river incision, lithological heterogeneity and climate at regional to continental scales. "

We will also add the following paragraph to clarify the objectives of the paper:

"Based on current limitations, we formulate two main objectives for this paper: we want (i) to assess the impact of lithological heterogeneity on river incision and (ii) to unravel the role of allogenic (spatial and/or temporal runoff variability) versus autogenic (incision thresholds) controls on river incision. We develop and evaluate our approach in the southern Ecuadorian Andes where detailed lithological information is available as well as a database of CRN-derived denudation rates (Vanacker et al., 2007, 2015)..

In the following sections, we first describe the study area, characterize the lithological configuration by developing a lithological erodibility index and compile a database to represent runoff variability. Second, we present the methods and assumptions used for calibrating and simulating river incision. In a third section, the modelling results are presented: we start by evaluating the impact of lithological heterogeneity on river incision rates using an area-based river incision model (A-SPM). We then evaluate to what extent the variability in denudation rates can be explained by spatial and/or temporal runoff variability and the existence of incision thresholds using the R-SPM and ST-SPM. Note that the goal of using R-SPM and ST-SPM models is not to improve the statistical explanatory power of the A-SPM but rather to get insights in the potential drivers of incision variability which are otherwise lumped in the parameters of the A-SPM. In a final section, we discuss our findings, highlight the implications of our work and discuss further perspectives. "

In the discussion, we added:

"Model performance of the ST-SPM equals the performance of an empirical A-SPM with a slope exponent >>1 (Figure 9). Our interpretation is that (i) spatial variations in runoff and (ii) the incision thresholds are the causes of an observed non-linear relation between ksn and ECRN. With a seemingly equal model performance, one could wonder what the benefit of the more complex ST-SPM model is over a simple, non-linear A-SPM. The aim of using a ST-SPM is however beyond fitting observed denudation rates: we want to identify to what extent the system is forced by internal allogenic dynamics such as the presence of incision thresholds or external autogenic forces such as runoff variability. Use of the ST-SPM illustrated that both processes can be accounted for in a quantitative way so that future studies can explicitly consider their role when reconstructing past landscape response to external perturbations (e.g. climate change)."

To Further clarify and stress this, we also adjusted the tile:

**"Parameterization of river incision models requires accounting for environmental heterogeneity: insights from the tropical Andes"**

I think the introduction needs to be revised somewhat to better reflect the results presented in the paper. The abstract does a better job of communicating the essence of the paper. Generally, the manuscript is very heavy on the methodology and too light on the discussion of the results and why these results matter.

We agree with the reviewer that we can organize our introduction somewhat better. As suggested in the line specific comments, we will also add some additional sentences throughout the manuscript to guide the reader better through the paper and to maintain a good flow in general. Our updated paper will be reorganised using the following section headers: 1.

1. Introduction

- 1.1.Background1.2.River incision models1.2.1.Area-based Stream Power Model1.2.2.Stochastic-Threshold Stream Power Model1.2.3.Runoff-based Stream Power Model2.Study area2.1.Geology
- 2.1.1. Tectonics and geomorphic setting
- 2.1.1. Lithological strength
- 2.2. CRN-derived denudation rates
- 2.3. River morphology
- 2.4. Runoff variability
- 2.4.1. Spatial runoff patterns
- 2.4.2. Frequency magnitude distribution of orographic discharges
- 3. Methods
- 3.1. CRN-derived denudation rates to calibrate river incision
- 3.2. River incision models
- 3.3. Optimization of model parameters
- 4. Comparing model results with CRN-derived denudation rates
- 4.1. Area-based stream power model
- 4.2. Runoff-based and Stochastic-Threshold Stream Power Models
- 4.2.1. Runoff-based SPM (R-SPM)
- 4.2.2. Stochastic-Threshold SPM (ST-SPM)
- 5. Discussion
- 5.1. Are CRN-derived denudation rates representative for long term river incision processes?
- 5.1.1. Equilibrium between river incision and hillslope denudation
- 5.1.2. Integration timescales of ECRN and ksn
- 5.2. Environmental control on long term river incision rates
- 5.2.1. Geology
- 5.2.2. Rainfall
- 6. Conclusions
- 7. References

We will also expand the result section and remove some of the methodology sections where possible. We will keep the section on the lithology since we think this part is necessary for the paper.

I also think the authors sometimes overreach on the significance of some results.

In the context of the clarified focus of the paper, discussed before, we will frame the results more clearly and mention the limitations explicitly.

It seemed like a long slog through the methodology section with many figures that did not seem terribly relevant OR were uninterpretable (Figures 3,5,7,8,9,11,13). Not all of these need to be relegated to Supplementary Material, but it would be helpful if some of them were and the important figures referenced more prominently in the text.

We will reduce the number of figures to 9, by merging some and moving others to the SI. We will remove the figure on PGA as suggested by the reviewer. In the updated version of the manuscript, we will also provide more details in the subscripts of the figures to make them understandable and readable as stand-alone objects.

I often felt like I had to hunt down the authors motivation for a methodology or intuit the reasons why results were significant. The authors need to be clearer throughout the manuscript on both of these points.

**Noted, we will revise.**

With some substantial improvements to this manuscript, particularly in cutting down the methodology section and refining and expanding the results section, I think it can be published as a valuable contribution to the geomorphology community.

We appreciate that the reviewer sees our work as a valuable contribution to the community. As mentioned before, we will try to cut down the methodology section where possible. Two essential parts of this paper – the high-resolution hydrological product and the lithological erodibility index will be kept in the main part of the manuscript, although we will cut down the text and move non-essential methodological aspects to the supplementary materials.

We will streamline the results section by consistently documenting all model parameters in one table (Table 4 rather than table 4 and 5) and will consistently refer to the scenarios as documented in the table. For the sake of clarity, we will present the model fits of all the scenarios in the supplementary information. Moreover, we realized that the discussion section could benefit from an additional graph reporting the overall model performance of the different models and will include this new figure (see below) in the revised document.

Additional figure: Comparison of model performance of four selected river incision models. (a) Nash Sutcliffe model efficiency (*NS*) for different model scenarios, without (grey bars) or with (red bars) considering lithological heterogeneity. (b) shows the corresponding Model Error (*ME*). The A-SPM model scenario corresponds to the Area-Based Stream Power Model (cf. Figure 7). It performs well when lithological heterogeneity is considered and all parameters are freely calibrated, resulting in an slope-steepness exponent (n; cf. Eq. 1) of 1.62 (for a full overview of model parameters, see Table 4). However, for an A-SPM scenario where n is fixed to the theoretically derived value of 1, the model performance strongly deteriorates (see main text). R-SPM represents a model scenario that explicitly incorporating runoff variability (cf. Figure 8a). The ST-SPM scenario also includes an incision threshold (cf. Figure 8b). Both scenarios perform well when n is fixed to 1 and when considering lithological heterogeneity. Overall, the best model performance (highest *NS* and smallest *ME*) is obtained under the ST-SPM scenario where lithological and runoff variability, as well as river incision thresholds are considered.*I have many specific comments on science issues and several technical corrections that are included in an annotated PDF that I will attach*.

We will address all the specific comments in the revised version of the manuscript.

**Line specific comments**

Line 16: 'enable to assess': typo

Where is the typo? Sentence rephrased.

Line 18 'variability of rock strength and its resistance to incision': wc

Isolating the role of rainfall variability remains difficult in natural environments, in part because environmental controls on river incision such as lithological heterogeneity are poorly constrained

Line 22 'Using 10Be catchment-wide erosion rates, meteorological and hydrological data, as well as data on bedrock erodibility, we provide quantitative constraints on the importance of rainfall variability and lithological variations': main point of the paper That is right...

Line 29 wc (word choice, reconsider) Research on how rainfall variability and tectonic forcing interact to make a landscape evolve over time has long been limited by the lack of techniques that measure erosion rates over sufficiently long timespans

Line 64 several small grammar mistakes: subject verb agreement Noted, we will revise.

Line 88: spelling We will write a new 'objectives' paragraph (see above)

Line 124 this needs to broken up into completely separate equations or at least labeled 4a, 4b, 4c. Psi is not defined in words and needs to be, as the threshold parameter. Noted, we will revise.

Line 127 a little confusing here as I was looking for the second component in equation 4. starting new paragraph should solve the problem Noted, we will revise.

Line 142. this is a big assumption. What needs to happen to make this true? some additional info from discussion can be moved up here. Good suggestion, we will add a paragraph with assumptions and revisit them in the discussion.

Line 152 model set 1: trad stream power That's right. We will name this one A-SPM consistently

Line156 second set of model runs with R-SPM. Above, should describe sets of model runs in the same order as incision models

In the theoretical section we first describe the ST-SPM because the R-SPM is a simplification of the ST-SPM. In the result section however, we believe it makes more sense to first present the R-SPM because it assumes constant *k* values and no thresholds. By presenting R-SPM first and then moving on to ST-SPM we gradually add layers of complexity which we find easier to navigate the reader through the result section.

Line 179: I see values of k (little k, no subscript) in table 2, but I don't see where it comes into the incision models. This needs to explained and clarifed. There are many parameters that are some version of big or little k with subscripts, superscripts, exponents, etc. Noted, we will revise.

Line 184: I think the paper would flow better if the organization was like this:

- 1. Introduction
- 2. all study area section
- 3. Explanation of River Incision models
- 4.1 Model runs using river incision models
- 4.2 Optimization of model parameters.

Good suggestion, see structure of updated manuscript as a response to general remark before.

Line 191 It doesn't seem like all of this information is necessary to arrive at the critical information in the final sentence of the paragraph.

We removed part of this section.

Line 206 possible to include a MAP map of the study area? That would be useful given the focus on characterizing the impacts of rainfall variability. If not feasible, at least give a value for MAP on the western slopes as well.

MAP is represented in Figure 6 of the updated manuscript, we will clarify.

Line 212 At some point in the paper (and maybe it's coming later), I would like to see a summary of the catchment erosion rates from these 30 sub-basins.

The erosion rates are given in Table 2. We will increase the size of the labels in Figure 1 to enhance clarity.

Line 227 Are these important for supporting the results/interpretations of this paper? Seems out of place to mention them here.

We suggest keeping the reference to these plots as they are key to understand the discussion on transient incision pulses (in the discussion section of the paper). We will however, move these lines to the new section 'River morphology' where they will be better in place.

Line 239 it would be helpful to readers to explain at the beginning of section 4.4 that the reason you do all of this is to get this regional kw value. Noted, we will revise.

Line 244 why is 45 stations with 10 years of data not enough? are they all clustered together?

Indeed, most of them are in the centre of the basin and do not cover the catchments where CRN derived erosion is measured (shown on Figure 1). We will clarify.

Line 248 this is bout 28 km resolution. pretty coarse.

While daily temporal resolution is really fine resolution to drive models that evolve over thousands of years.

That is right, therefore we develop a HR product by downscaling the 0.25° WaterGAP3 data. See also next reply.

Line 252 I don't easily grasp the relevance of this section, especially the second half of the paragraph, starting on line 248. What needs to happen in this paragraph is a more succinct explanation of the data sets used to get a pdf of daily runoff and more importantly, why using these data sets is an improvement on the data from the monitoring networks on the ground. We will shorten and rephrase this paragraph as:

"To estimate runoff variability for all 30 sub catchments, we use hydrological data derived in the framework of the Earth2Observe Water Resource Reanalysis project (WRR2; Schellekens et al., 2017) available from 1979 to 2014. Specifically, we use the hydrological data calculated with the global water model WaterGAP3 (Water – Global Assessment and Prognosis: Alcamo et al., 2003; Döll et al., 2003) at a spatial resolution of 0.25° and a daily temporal resolution (earth2observe.eu). In the following paragraphs, we explain how we derive (i) a high-resolution runoff map by spatially downscaling this coarse data and (ii) catchment-specific magnitude frequency distributions of discharge ( $pdf_Q^*$ ) characterising the temporal variability of runoff."

Line 255 nice intro and motivation for methodology here. But, before you get into the detailed explanation of methods, refer readers to figure 6 so they get a visualization of where you're going and why you do this.

**Thanks.**

We will point the readers to figures 5 and 6:

"The procedure consisted of the following steps and is presented in Figures 5 and 6:"

Line 281 This sentence unnecessarily confusing. Use more words to explain. this section needs an introductory sentence to orient readers.

We will resolve by adding the following text:

"Runoff variability is typically casted in terms of spatial runoff variability (section 2.4.1). However, also the temporal pattern of runoff might influence river incision and is typically represented by discharge magnitude frequency distributions. Constraining the shape of these distributions is important, because the number of large storm events determine the frequency by which thresholds for river incision to occur are exceeded (see section 1.2.2 and references therein). Line 285 here little k is finally defined. this needs to happen earlier where it is first mentioned. Noted, we will revise.

Line 294 how important are daily variations in discharge over 9 million years of uplift and erosion? Good point, we will mention this earlier in the assumption section coming with the river incision models and revisit the issue in the discussion section of the paper.

Line 297 this is all fine and good, thorough work, but the summary/motivation of why you do this needs to be at the beginning of the paragraph. otherwise makes for very heavy reading. Agreed, see reply above.

Line 300 is this section necessary? Does it really contribute to the main goal of the paper, which I understand to be evaluating the role of rainfall variability on incision rates. this sections feels like overkill. I recommend moving to supplementary materials.

As explained earlier, we do believe this section is critical given the importance of lithological heterogeneity in controlling river incision rates. Therefore, we will keep it in the methodology section.

Or at least the seismicity section can go to supp mat. Agreed, we kicked it out.

the lithological strength section should actually stay as it's very important later in the paper. Indeed... Should now also be clear to the readers when they arrive at this point, given the enhanced focus on lithology in the abstract/intro

Line 317 where are these data of measured uniaxial compression strength? OK, I have found it now, but this section is confusing. it needs to be more clear and use more words to explain We will explicitly mention that the uniaxial compressive strength data can be found in Table S4 to enhance clarity.

Line 319 this part definitely seems irrelevant to the main focus of the paper. There are so many other things already going on, this just feels like a distraction. Unless seismic activity is really playing a huge role, in which case maybe the focus of the paper should be on that.

Agreed, we kicked it out.

Line 327 reference figure 12, not table 4.

We will redo the figure numbers in the revised version of the paper and point the reader to both the table and the figure.

Line 331 more explanation and description of figure 12 would be helpful here before launching into another lengthy description of another methodology.

Agreed, we describe all scenarios now in more detail.

Line 336 what about Bayes factors of 1.06 vs 1400 tells us that the data fit a model with variable erodibility better? Needs more explanation. See comment before.

Line 346 coming back to an earlier comment from near the beginning of the manuscript, it seems the focus of this paper is equally on the effects of spatial variation in both erodibility and runoff. this is not clear/emphasized in the abstract and introduction.

Agreed, we will resolve this by rewriting the abstract, and objectives of the paper. See comments before.

Line 351 spell out model names for section title.

Noted, we will resolve. To clarify, we will also break up this section in two subsections:

- 4.2. Runoff-based and Stochastic-Threshold Stream Power Models
- 4.2.1. Runoff-based SPM (R-SPM)
- 4.2.2. Stochastic-Threshold SPM (ST-SPM)

Line 353 nice explanation of what just happened and what will happen next. manuscript needs more of this in places.

Thanks, and agreed, we will resolve.

Line 360 also good emphasis. Thanks

Line 362 this under prediction/over prediction trend is not obvious to me. Looks to me like the observational data is scattered somewhat evenly about the modeled data line. If the Nash Sutcliff number tells us that there is under/over prediction, then explain how that happens. Otherwise, I think such a claim is not supported.

**We will rewrite this paragraph**

Line 369 these two scenarios evaluate role of little k. True, we will clarify.

Line 372 little k apparently not so important.

Correct, we will stress this as: "In scenario 5, k is fixed to the average value for all catchments (k = 1.01) whereas in scenario 6, k is set to the catchment specific values as listed in Table 2. Both scenarios (5 and 6) perform well with an NS value equalling 0.71 indicating that temporal runoff variability (k) is not influencing model performance."

Line 375 here finally runoff variability is evaluated. No runoff is evaluated before. This will be clearer in the revised result section.

Line 378 this is misleading, as scenario 7 performs equally well! We will explicitly mention that both scenarios perform equally well.

Line 379 what's the significance of these lower threshold values? We will explain and frame it with some data from literature.

Line 380 I agree that the model vs. measured erosion rates look better in 14b compared to 14a. But it's an over-reach to say that ST-SPM "correctly" predicts low erosion rates. There's still a good bit of scatter and error in model vs. measured ero rates. Just modulate the word choice a little here.

Noted, we will revise.

Line 386 some of this context would be helpful earlier in the paper, e.g. near line 142 or section 4.2

Noted, we will revise by pointing the readers to this section

Line 401 how does this relate to the data presented? where do you suspect over/under estimation of ero rates from Be10?

This will be discussed in the next paragraph.

Line 406 these ids are hard to see and find. note that they are in teh northern section of field area and also refer readers to figure 2, can see these areas are steeper. All good suggestions, we will do so. Line 409 confusing. is the variability in agreement due to differences in drainage area in each catchment or to over/under estimation of CRN erosion rates? This needs to be more precisely worded, as the following sentences make clear.

We rewrote this paragraph:

"Longitudinal profiles of rivers draining to the knickzone in the Paute catchment show marked knickpoints. This is particularly evident in catchments 9-16 (Figure 1) where ksn values are high (Figure 2) and knickpoints appear in the longitudinal profiles (Figures S3 and S4). Simulated erosion rates for some of these catchments deviate from CRN-derived denudation rates (Figure 8.b, ID's 13 14 and 16) whereas for others (e.g. ID's 9 and 11), predictions from the Stochastic-Threshold river incision model show a good agreement with ECRN data. For catchments with a sufficiently large drainage area, modelled incision rates correspond well with ECRN (ID's 9 and 11 being both ca. 700 km2), most likely because the mechanisms that potentially cause overestimation and underestimation cancel each other out at this scale. For smaller catchments (ID's 8;13;14 and 16 all being < 12 km2) there is a discrepancy between simulated river incision rates and ECRN."

Line 427 good point that will have been on the mind of many readers through out the paper. maybe acknowledge this timescale mismatch earlier.

Good suggestion, we will do so by adding a paragraph to the methods section: (3.1)

"The use of CRN-derived denudation rates to calibrate river incision relies on three main assumptions, summarized by Scherler et al. (2017). A first assumption is that the catchment wide denudation rates derived from CRN are representative for long term fluvial incision. Positive correlations between river steepness, ksn and CRN-derived denudation rates support this assumption (Vanacker et al., 2015), except for very small catchments where CRN-derived denudation rates are sensitive to the occurrence of deep-seated landslides. A second assumption is that runoff and rock uplift are uniform within the individual catchments. Given the size of the studied basins, this assumption seems to be reasonable. A third assumption, in particular when using the process-based R-SPM and ST-SPM, is that the runoff data, used to calibrate the incision parameters is representative over the time span which CRN data integrate (1-100 kyr). This is a challenging assumption, given the contemporaneous nature of the available hydrological data. While spatial patterns of runoff, mainly controlled by orographic precipitation, could be assumed broadly similar over the integration time of CRN-derived denudation, this is not necessarily true for the temporal variation in runoff. We will revisit the validity and implications of these three assumptions in the discussion section of this paper. "

Line 437 a bit much detail at this point. Agreed, we removed part of this paragraph. Line 444 I must have completely missed this point. Refer readers back to the relevant model runs/figures.

Noted, we will revise.

Line 447 allowed us Noted, we will revise.

Line 449 given what you just said, now useful are more advanced methods likely to be? where would they be useful? We will discuss this by adding a sentence.

Line 456 this seems to be the major, clear finding of this work. We will revise the text to clarify the main findings (and also updated the title of the paper)

Line 467 if this is your conclusion, recommend removing earlier discussion of seismicity. Agreed, we will revise.

Line 475 this is also a good and significant point that could be highlighted more prominently. We will do so and mention it in the abstract of the paper.

Line 484 enables us Noted, we will revise.

Line 485 am i missing something? this scenario does NOT include variation in runoff. Hmm, we do not exactly know what the reviewer means here. But we will rephrase the sentence and the paragraph in general.

Line 486 above error/confusion makes it hard to evaluate this very important claim. Agreed, we will rephrase this sentence and the paragraph in general.

Line 494 OK, good point here.

Line 495 yes, but the numbers are only slightly higher for scenario 6 vs scenario 4. How significant is this seemingly small difference?

We will rewrite this paragraph, as well as the paragraph in the results section dealing with these scenarios. In the results section we will add a sentence explicitly mentioning the similarity between these scenarios:

"Note that differences in model performance between R-SPM scenario 2 and ST-SPM scenarios 5-8 are existent but not very pronounced. To evaluate the significance of these differences, our analysis should be repeated on larger datasets capturing a wider variability in denudation rates and hydrology.

Line 503 i would say that mainly spatial variation in rock erodibility controls river incision patterns. I have to say, I'm not convinced of that rainfall variability matters a huge amount from the data presented here.

We rephrased this paragraph. See also replies above.

"Our finding that spatial patterns in precipitation control river incision patterns corroborate findings in the Himalaya (Scherler et al., 2017) and in the Andes (Sorensen and Yanites, 2019). Sorensen and Yanites (2019) evaluated the role of latitudinal rainfall variability in the Andes on erosional efficiency using a set of numerical landscape evolution model runs. They show that erosion efficiency in tropical climates at low latitudes, where the Paute basin is located, is well captured by the spatial pattern of mean annual precipitation and thus runoff. At higher latitudes (25-50°) where storms are less frequent but still very intense, mean annual precipitation decreases but erosivity is still high due to the intensity of storms (Sorensen and Yanites, 2019). At these latitudes, the spatial variations in storm magnitude are therefore more likely to be reflected in river erosivity and thus catchment mean denudation rates than in the Ecuadorian Andes."

Line 518 medium? anyway, medium is not a great word choice to describe basin size. Noted, we will revise.

Line 527 But the simplest version of the model, A-SPM does almost as good of a job! R^2=0.73, NS=0.73! You must explain this and justify why variable R actually matters! See comments before. We rephrased this paragraph as:

"In order to account for rock strength variability, which is for the Paute basin mainly ascribed to variations in lithological strength in the study area, we propose the use of an empirical lithological strength index that is based on lithology and age of lithostratigraphic units. Including lithological variability in the models increases the correlation between river steepness and denudation rates and reveals a non-linear relation, which we seek to explain using a stochastic-threshold SPM (ST-SPM). Using a downscaled version of a state-of-the-art hydrological reanalysis dataset, we show that the combination of spatially varying runoff and incision thresholds explains the observed, non-linear relationship. We do not detect, however, an impact of temporal discharge distributions on river incision. We attribute this lack to the integration time of CRN data and response times of river longitudinal profiles which extend beyond timescales at which discharge distributions can be assumed to be stationary."

Line 535 I think this conclusion is fair that this study shows potential, but more research is still needed for definitive answers about R variability.

We will keep this message in the revised version of the paper.

**Lithology and orographic precipitation control Parameterization of river**

incision immodels requires accounting for environmental heterogeneity: insights from the tropical Andes

Benjamin Campfortsa,b,c, Veerle <del>Vanackere Vanackerd</del>, Frédéric HermaneHermane, Matthias <del>Vanmaerckee Vanmaerckef</del>,

5 Wolfgang SchwanghartfSchwanghartg, Gustavo E. Tenoriob,gTenorioh,g Patrick WillemshWillemsh Gerard Goversh

a Helmholtz Centre Potsdam, GFZ German Research Centre for Geosciences, Potsdam, Germany b Institute for Arctic and Alpine Research, University of Colorado at Boulder, Boulder, CO, USA a Research Foundation Flanders (FWO), Egmontstraat 5, 1000 Brussels, Belgium

10 b-Department of Earth and Environmental Sciences, KU Leuven, Celestijnenlaan 200E, 3001 Heverlee, Belgium ed Earth and Life Institute, Georges Lemaître Centre for Earth and Climate Research, University of Louvain, Place Louis

Pasteur 3, 1348 Louvain-la-Neuve, Belgium

de Institute of Earth Surface Dynamics, University of Lausanne, CH-1015 Lausanne, Switzerland

- e Université de Liège, Département de Géographie University of Liege, UR SPHERES, Departement of Geography, Clos
   Mercator 3, 4000 Liège, Belgium
- fe Institute of Earth and Environmental SciencesScience and Geography, University of Potsdam, Germany
   fe Facultad de Ciencias Agropecuarias, Universidad de Cuenca, Campus Yanuncay, Cuenca, Ecuador
   h Department of Earth and Environmental Sciences, KU Leuven, Celestijnenlaan 200E, 3001 Leuven, Belgium
   L Department of Civil Engineering Hydraulics Section, KU Leuven, Kasteelpark 40 box 2448, 3001 Leuven, Belgium
- 20 Correspondence to: Benjamin Campforts (benjamin.campforts@kuleuven.begfz-potsdam.de)

Abstract. Process-based geomorphic transport laws enable Landscape evolution models can be used to assess the impact of rainfall variability on bedrock river incision over geologicalmillennial timescales. However, isolating the role of rainfall variability on erosion-remains difficult in natural environments, in part because the variability of rock strength and its resistance to incisionenvironmental controls on river incision such as lithological heterogeneity are poorly constrained. HereIn this study, we explore spatial differences in the rate of bedrock river incision in the tropical Andes. The Ecuadorian Andes are characterized by strongusing three different stream power models. A pronounced rainfall gradientsgradient due to orographic precipitation sourced in the Amazon basin. In addition, the tectonic configuration has generatedand a profoundhigh lithological heterogeneity. The enable us to explore the relative roleroles of either these controls in modulating river incision on millennial time scales, however, remains unclear. Using 10Be catchment wide erosion rates, meteorological and

- hydrological data, as well as data on bedrock erodibility, we provide quantitative constraints on. First, we use an area-based stream power model to scrutinize the role of lithological heterogeneity on river incision rates. We show that lithological heterogeneity is key to predicting spatial patterns of incision rates. Accounting for lithological heterogeneity reveals a nonlinear relationship between river steepness, a proxy for river incision, and cosmogenic radio nuclide (CRN) derived
- 35 denudation rates. Second, we explore this nonlinearity using runoff-based and stochastic-threshold stream power models, combined with a state-of-the-art hydrological dataset to calculate spatial and temporal runoff variability. Statistical modelling suggests that the importance of rainfall variability and lithological variations. Explicit incorporation of rock erodibility in river incision models predicated onnon-linear relationship between river steepness and denudation rates can be attributed to a spatial runoff gradient and incision thresholds. Our findings have two main implications for the stream power equation enables.

| Formatted: Font: +Body (Calibri), 12 pt, Font color:
Black                              |
|---------------------------------------------------------------------------------------------------|
| Formatted: Left, Line spacing: Multiple 1,08 li                                                   |
| Formatted: Numbering: Restart each section                                                        |
| Formatted: Font: +Body (Calibri), 12 pt, Font color:
Black                              |
| Formatted: Font: +Body (Calibri), 12 pt, Not Bold, Font color: Custom Color(RGB(33,33,33)) |
| Formatted: Font: 10 pt                                                                            |
| Formatted: Font: 10 pt, Superscript                                                               |
| Formatted: Font: 10 pt                                                                            |
| Formatted: Font: 10 pt, Italic                                                                    |
| Formatted: Left, Space After: 0 pt, Line spacing:
Multiple 1,15 li                             |
| Formatted: Line spacing: Multiple 1,15 li                                                         |
| Formatted: Line spacing: Multiple 1,15 li                                                         |
| Formatted: English (United Kingdom)                                                               |
| Formatted: English (United Kingdom)                                                               |
| Formatted: Space After: 8 pt, Line spacing: Multiple 1,15 li                                      |
| Formatted: Superscript                                                                            |
| Formatted: Normal, Line spacing: Multiple 1,15 li                                                 |
| Formatted: English (United Kingdom)                                                               |
|                                                                                                   |

[revised manuscript text omitted]
 [L3t-1] by mean-annual discharge  $\overline{Q}$  [L3t-1],  $k_e$  [L2.5  $\mathbb{T}^3 \mathbb{T}^2 \mathbb{m}^{-1.5}$ ] is the erosional efficiency constant,  $\overline{R}$  [L t-1] is the mean annual runoff, ais the shear stress exponent reflecting the nature of the incision process (Whipple et al., 2000),  $\psi$  is the threshold term [L t-1], and  $k_i$ ,  $k_w$ ,  $\alpha$ ,  $\beta$ ,  $\varphi_{i\alpha}$  and  $\varphi_{ib}$  are channel hydraulic parameters described in Table 1. The Table 1.

In a second component derives tep, long term river erosionincision is calculated by multiplying-the instantaneous river incision, *I*, calculated for a discharge of a given magnitude ( $Q^*$ ) with the probability for that discharge to occur ( $pdf(Q^*)$ , see section 5.1.2)\*)) and subsequently integrating this product over the range of possible discharge events specific to the studied timescale (DiBiase and Whipple, 2011; Lague et al., 2005; Scherler et al., 2017; Tucker and Bras, 2000; Tucker and Hancock, 2010):

$$E = \int_{Q_c^*}^{Q_m^*} I(Q^*) \, p df(Q^*) dQ^* \tag{5}$$

in which  $Q_c^*$  is the minimum normalized discharge which is required to exceed the critical shear stress ( $\tau_c$ ) and  $Q_m^*$  is the maximum possible normalized discharge over the time considered.

**1.2.3. Runoff-based Stream Power Model**

A third river incision model further discussed inderived from the paperST-SPM, is the Runoffrunoff-based SPM (R-210 SPM). The R-SPM shares its derivation withis similar to the ST-SPM, but assumes riverthat the incision thresholds to beare negligible ( $\psi = 0$ ) and that discharge to beis constant over time ( $Q^* = 1$ ), simplifying Eq. (5)5 to:

Formatted Table

| Formatted: Font: Italic                |
|----------------------------------------|
| Formatted: Font: Italic                |
| Formatted: Indent: First line: 1,27 cm |
| Formatted: Font color: Auto            |

 $E = K k_s^n$

(8)

In the following sections, we first describe the study area, characterize the lithological configuration by developing a lithological erodibility index and compile a database to represent runoff variability. Second, we present the methods and assumptions used for calibrating and simulating river incision. In a third section, the modelling results are presented: we start by evaluating the impact of lithological heterogeneity on river incision rates using an area-based river incision model (A-SPM). We then evaluate to what extent the variability in denudation rates can be explained by spatial and/or temporal runoff variability and the existence of incision thresholds using the R-SPM and ST-SPM. In a final section, we discuss our findings, highlight the implications of our work and discuss further perspectives.

220 3.1. Methods

**3.1.1.1. Optimization of model parameters**

The presented forms of the stream power model-all depend on river steepness,  $k_{m}$ , known to correlate well with  $E_{CRN}$ (DiBiase et al., 2010; Ouimet et al., 2009; Scherler et al., 2017; Vanacker et al., 2015), Moreover,  $E_{CRN}$  integrate over timespans that average out the episodic nature of erosion and over spatial extents large enough to average out the stochastic nature of hillslope processes. Moreover, if we assume that river incision occurs at rates of catchment-wide denudation,  $E_{CRN}$ can be used to constrain models of river incision (cfr. DiBiase and Whipple, 2011; Scherler et al., 2017).

To optimize model parameters, we maximize the Nash Sutcliff model efficiency (*NS*, Nash and Suteliffe, 1970) between observed erosion (*O*) and modelled river incision (*M*):

$$NS = 1 - \frac{\sum_{i=1}^{i=n_{i}} (\theta_{i} - M_{i})^{2}}{(\theta_{i} - \bar{\theta})^{2}}$$
(7)

where *nb* is the number of ECRN samples. The *NS* coefficient ranges between -∞ and 1 where 1 indicates optimal model performance explaining 100 % of the data variance. When *NS* = 0, the model is as good a predictor as the mean of the observed data. When *NS* <= 0; model performance is unacceptably low. The *NS*-coefficient has been developed in the framework of hydrological modelling but has been applied in wide range of geomorphologic studies (e.g. Jelinski et al., 2019; Nearing et al., 2011).

**3.2. River incision models**

235

240

as:

225

In a first set of model runs, we evaluate the performance of the A-SPM in predicting  $E_{CRN}$  rates. To account for rock strength variability Eq. (2) is rewritten as:

$$E = k_a \overline{L_E} k_{sn}^{-n}$$

where  $k_{\alpha}$  (L1-2mt-1) is the crossional efficiency parameter and  $\overline{L_{L}}$  is a dimensionless catchment mean lithological crodibility value.

[revised manuscript text omitted]

(15)

where *nb* is the number of ECRN data points, Oi are the catchment specific measured ECRN denudation rates, Mi represents the catchment specific standard deviation on ECRN. The advantage of the ME is that it explicitly incorporates the error on the analytical data (ECRN) by weighing the model error with the analytical error. However, errors on CRN data are heteroscedastic: they systematically increase with increasing denudation rates. Although the *ME* thus provides a good metric to evaluate overall model performance, the metric is not well suited to optimize model parameters in an optimization procedure: too much weight will be given on optimization of the model in the lower regime of the denudation spectrum where measured errors on ECRN are low whereas higher measured ECRN data will not be approximated well because of large associated errors. To compensate for the effect of heteroscedasticity we rescale values Oi, Mi and Ei using a logarithm with base 10 when calculating ME (Herman et al., 2015). In this paper, ME will be used to evaluate model performance, but not to optimize model parameters.

A second metric is the coefficient of determination, *R*2. Contrary to *ME*, *R*2 evaluates the explained variance of the model giving all observations the same weight, regardless their analytical error. However, when model parameters result in an offset between simulated and observed data (i.e. the intercept of the fit), this can still result in a high *R*2.

We therefore use the Nash Sutcliff model efficiency to optimize model parameters (NS, Nash and Sutcliffe, 1970) :

$$NS = 1 - \frac{\sum_{i=1}^{i=nb} (O_i - M_i)^2}{(O_i - \bar{O})^2}$$
(16)

570

575

580

The NS coefficient ranges between  $-\infty$  and 1 where 1 indicates optimal model performance explaining 100 % of the data variance. When NS = 0, the model is as good a predictor as the mean of the observed data. When NS <= 0; model performance is unacceptably low. The NS-coefficient has been developed in the framework of hydrological modelling but has been applied in wide range of geomorphologic studies (e.g. Jelinski et al., 2019; Nearing et al., 2011).

**4. derived erosion rates (ECRN). When erodibility is Comparing model results with CRN-derived denudation rates**

In the following sections, we compare simulated erosion rates, obtained with the river incision models presented in Eq. 11 – Eq. 13 with measured CRN-derived denudation rates. We start with the use of the A-SPM (Eq. 11) to evaluate the extent to which lithological variability controls denudation rates. Once the impact of lithological heterogeneity on river incision is clarified, we evaluate whether runoff variability and incision thresholds can explain variations in ECRN.derived denudation rates. To this end, two process-based river incision models are evaluated (the R-SPM and ST-SPM, presented in Eq. 12 and Eq. 13 respectively). Optimized parameters and model performance of all model scenarios are listed in Table 4. Best fit results of a selected number of model runs are presented in Figure 7 and Figure 8. An overview of model fits for all the scenarios listed in Table 4 is given in Figures S8, S9 and S10.

**4.1. Area-based stream power model**

In a first set of model runs we evaluate the use of an Area-Based Stream Power Model (A-SPM) to explain observed variations in CRN-derived denudation rates ( $E_{CRN}$ ). We optimize river incision parameters for four scenarios (Table 4: A-SPM scenario's 1 – 4): in the first two scenarios, the slope exponent, *n* is left as a free parameter. In the second two scenarios, the slope parameter is fixed to unity (*n* = 1).

In A-SPM scenario 1 (Table 4, Figure 7.a), we assume a spatially uniform, long term river incision (*E*) is a power function of the normalized river steepness  $k_{sm}$  scaled by an erodibility ( $\overline{L_E}$  fixed to 1 in Eq. 11) and leave the erosion efficiency coefficient (*K'*). By optimizing-) and the slope parameter *n* as free parameters during model optimization. The optimized fit between Esimulated erosion (*E*, Eq. 2) and ECRN, *K'* and *n* are constrained is shown in Figure 7.a. The fit is surrounded by a lot of data scatter resulting in a *NS* model efficiency of 0.5, a *R2* of 0.5, a *ME* of 3.25 and an optimized value for *n* of 1.06 (Figure 12.a, Table 4). When includingoptimized values for *K'* and *n* of respectively 0.57 m0.1s-1 and 1.12. The fit between simulated and measured denudation rates hints to the existence of a correlation between ECRN and river incision rates. The fit shown in in Figure 7.a, shows that modelled erosion rates for catchments with a low mean erodibility index (= high resistance to erosion) are mostly overpredicted (plotting below the 1:1 line) whereas modelled erosion rates of catchments with a high

erodibility index are mostly underpredicted (plotting above the 1:1 line).

In A-SPM scenario 2 (Table 4, Figure 7.b), we quantify the impact of varying lithology by using catchment specific mean-values for the lithological erodibility values ( $\overline{L_E}$ ), in Eq. 11) and leaving  $k_a$  and n as free optimization parameters. The optimized fit between simulated erosion (E, Eq. 11) and ECRN is shown in Figure 7.b. Optimization results in a NS model efficiency of 0.73, a  $R^2$  of 0.73, a ME of 2.23 and optimized values for  $k_a$  and n of respectively 0.07 m0.1s-1 and 1.64.

585 Considering lithological erodibility strongly increases (NS = 0.73) and the optimized value of *n* equals 1.63.

To evaluate whether including spatially varying erodibility values also increases the predictive power of the river incision model, we performed a linear Bayesian regression analysis between  $E_{CRN}$  and the simulated long term river erosion 19

*E*. Figure 13 shows that the posterior probability of linear regression coefficients close to one is higher and with less spread when considering spatially varying lithological erodibility values. Moreover, when *E* is only a function of  $k_{sm}$ , the Bayes factor equals 1.06, in comparison to a value of ca. 1400 when *E* is a function of both  $k_{sm}$  and  $\overline{L_E}$  (Table 4). This implies that a river incision model accounting for variable erodibility values is supported by the data (Jeffreys, 1998).

reduces data scatter surrounding the fit. The importance of lithological strength in controlling the A-SPM and the  $k_{sn}$ -ECRN relation confirms that strong metamorphic and plutonic rocks erode at significantly-slower rates than lithologies which are less resistant to weathering such as sedimentaryvolcaniclastic deposits of loose volcanic mixtures. The empirical rock strength classificationerodibility index we developed appears to be provide an appropriate scaling of relative rock strength: analysis of residuals did not reveal any significant relation of residuals with lithology.

When using spatially variable, catchment specific lithological erodibility values ( $\overline{L_E}$ ) (Figure 12.(Figure 7.b), the *n* coefficient of the SPM is considerably larger than unity (n = 1.6364) and the  $k_{sn}$ -ECRN relationship becomes non-linear, corroborating earlier findings documented in e.g. Gasparini and Brandon (2011). While this may be due to the fundamental properties of river incision and erosion processes, the shape of the relation may also be affected by spatial covariates other than lithology. In the following sections, we will investigate whether this nonlinear  $k_{sn}$ -ECRN relationship can be explained by the presence of incision-thresholds, variations in runoff, or a combination of bothTo evaluate the impact of a variable *n* exponent on the performance of the empirical A-SPM, we executed two more model optimizations.

**605 6.2. R-SPM and ST-SPM**

In A-SPM scenario 3 (Table 4, Figure S8.c), we assume a spatially uniform lithology and erodibility ( $\overline{L_E}$  fixed to 1 in Eq. 11), fix *n* to 1 and only leave *K*' to be optimized as a free model parameter. With a *NS* model efficiency of 0.5, a  $R^2$  of 0.5, a *ME* of 3.2 and an optimized value for *K*' of 1.00 m0.1s-1, the model fit and performance is similar to the values obtained in scenario 1.

510 In A-SPM scenario 4 (shown in Table 4, Figure S8.d), lithological variability is considered (using catchment specific values for LE in Eq. 11), *n* is fixed to 1 K' is a free model parameter. With a *NS* model efficiency of 0.51, a R2 of 0.56, a *ME* of 3.05 and an optimized value for K' of 1.4 m0.1s-1, model performance is much lower than when leaving the slope exponent *n* as a free parameter (A-SPM scenario 2). This result shows that the apparent lack of a non-linear relationship between river steepness (*ksm*, representing river incision rates) and ECRN (scenario 1 and 2) can be explained by lithological heterogeneity which is masking the existence of such non-linear relationship. Once lithological variability is considered, a linear relationship with *n* =1 between *ksm* values and ECRN (this scenario, A-SPM 4) is performing less well than a river incision model where this relationship is non-linear (with *n*>>1).

**4.2. Runoff-based and Stochastic-Threshold Stream Power Models**

620

625

590

595

600

The previous analysis shows that the explanatory power of the A-SPM model, and therefore the  $k_{sn}$ -ECRN relationship, strongly improves when considering spatial variations in lithological erodibility-lithology. Moreover, when considering variations in lithological erodibility, river incision is found to be non-linearly dependent on the channel slope (*S*), with n = 1.63. In a next step we evaluate whether this non-linear relation can be explained by spatial and/or temporal rainfall variability and/or the existence of thresholds for river incision (Table 5)-(Table 4: R-SPM scenarios 1 - 2 and ST-SPM scenarios 1 - 8, Figure 8).

**4.2.1. Runoff-based SPM (R-SPM)**

In a first set of model runs, we evaluate the performance of the runoff-based Stream Power Model (R-SPM in combination with Eq. 12) to evaluate the role of spatially variable runoff using catchment specific values for mean runoff (Table 2). When (*R* derived from the Water GAP data, reported in Table 2 and shown in Figure 6).

[revised manuscript text omitted]

(incision thresholds) response is casted in the lumped empirical incision parameters. While the R-SPM and ST-SPM do not necessarily predict spatial patterns in observed  $E_{CRN}$  rates better than an A-SPM, they do enable to simulate the effect of runoff variability and incision thresholds and therefore provide an operational tool to simulate past and future climate changes. Note that differences in model performance between R-SPM scenario 2 and ST-SPM scenarios 5-8 are existent but not very pronounced. To evaluate the significance of these differences, our analysis should be repeated on larger datasets capturing a wider variability in denudation rates and hydrology.

**7.5. Discussion**

**7.1.5.1. Are CRN\_derived erosiondenudation rates representative for long term river incision processes?**

**7.1.1.5.1.1. Equilibrium between river incision and hillslope denudation**

710

730

700

705

Assuming an equilibrium betweenIn theory, rates of hillslope denudation equal rates of river incision and hillslope erosion theoretically holds forif landscapes which are either in a steady state or forif transient landscapes are characterized by rapid hillslope response (e.g. threshold hillslopes). Steady state landscapes can only be achieved under stable precipitationclimatic and tectonic settings that prevail over timescales exceeding several millions of years. Such configurationstability is rarely met in tectonically active regions where riverslandscapes continuously transmit newrespond to environmental perturbations to the upper parts of the catchment (Armitage et al., 2018; Bishop et al., 2005; Campforts and Govers, 2015).

The downstream reaches of the Paute catchment are a good example of a transient landscape where a major knickzone (Figure S3 is propagating upstream in the catchment resulting in steep threshold topography downstream of the knickzone (Figure S3 and Vanacker et al., 2015). Facing a sudden lowering of their base level after river rejuvenation, soil production and linear hillslope processes such as soil creep (Campforts et al., 2016; Vanacker et al., 2019) are not any longer able to catch up(Campforts et al., 2016) are not any longer in equilibrium with rapidly incising rivers (Fig. 15 in Hurst et al., 2012). In transient regionssteep topography, hillslopes may transiently evolve to their mechanically limited threshold slope where any further perturbation of threshold hillslopes will result in increased sediment delivery through mass wasting processes such as rockfall or landsliding (Bennett et al., 2016; Blöthe et al., 2015; Burbank et al., 1996; Larsen et al., 2010; Schwanghart et al., 2018), Given the stochasticerratic nature of landslides, not all threshold hillslopes will respond simultaneously to base level lowering depending on local variations in rock strength, hydrology and seismic activity (Broeckx et al., 2019). Therefore, catchments in transient regions might experience erosion in a broad range from moderate to high rates with similar

725 probabilities, land use and seismic activity (Broeckx et al., 2020; Guns and Vanacker, 2014). Therefore, catchments in transient landscapes might experience hillslope denudation with highly variable rates.

Thus, We argue that CRN-derived erosiondenudation rates mightin the Paute basin both overestimate and underestimate long term incision rates in these catchments. Overestimation resultsmay result from the occurrence of recent, deep-seated landslide events, that deliver sediments with low CRN concentration to rivers (Tofelde et al., 2018). Underestimation might, in turn, may occur if long-term hillslope lowering is accomplished by rare and large landslides characterized by the occurrence of rare, large events with awhose return period exceeding eriods exceed the integration time of CRN-derived erosiondenudation rates, (Niemi et al., 2005; Yanites and Tucker, 2010).

Longitudinal profiles of rivers draining to the knickzone in the Paute catchment show marked knickpoints ( $ID^{*}s$ . This is particularly evident in catchments 9-16 on Figure 1; (Figure 1) where  $k_{str}$  values are high (Figure 2) and knickpoints appear

735 in the longitudinal profiles (Figures S3 and S4). Figure 14.b shows that simulated Simulated erosion rates for some of these catchments deviate from CRN--derived erosiondenudation rates (Figure 8.b, ID's 13 14 and 16) whereas for others (e.g. ID's 9 and 11), predictions from the stochastic thresholdStochastic-Threshold river incision model show a good agreement with ECRN data.-We attribute this variability to differences in drainage area between these catchments. For catchments with a sufficiently large drainage area, modelled incision rates correspond well with ECRN (ID's 9 and 11 being both ca. 700 km2), 740 most likely because the mechanisms that potentially cause overestimation and underestimation cancel each other out at this scale. For smaller catchments (ID's-8;13;14 and 16 all being < 12 km2) there is a discrepancy between simulated river incision rates and ECRN.

Although river incision ratesmodels can be used to estimate general erosionsimulate denudation patterns in large transient catchments (>>(> 10 km2), there is a need to develop alternative approaches to simulate erosion rates in transient regions over different spatial scales. One such approach could be the explicit integration of including e.g. landslide mechanisms in long term landscape evolution models such as TTLEM (Campforts et al., 2017) or Landlab (Hobley et al., 2017) to capture the stochastic nature of these processes (Niemi et al., 2005; Yanites et al., 2009)(Hobley et al., 2017).

**7.1.2.5.1.2. Integration timescales of ECRN and ksn**

CRN concentrations in detrital sediments integrate over timescales dependent on the erosion rate of the catchment. For a rock density of 2.7 g cm-3, the integration time corresponds to the time required to erode ca. 60 cm of rock (Kirchner et al., 2001). ECRN in the Paute basin varies between 5 to 399 mm yr4 implying integration times ranging from ca. 1.5 to 175 ky. Topographical river profiles on the other hand are the outcome of the dynamic interplay between tectonics, lithology, rainfall variability and internal drainage reorganization over timescales well exceeding one million yearsOur analysis reveals the 755 potential role of temporal and spatial variations of rainfall in long term landscape evolution. Integration times of CRN-derived denudation rates measured in the Paute basin are in the order of 1.5-175 ky. In contrast, response times of longitudinal river profiles generally range from 0.25-2.5 Ma (Campforts et al., 2017; Goren et al., 2014; Wobus et al., 2006).

Thus, successful identification of a rainfall variability signal is only possible if the signal has been present during the integration timescale of both ECRN and km. Given the high sensitivity of extreme precipitation events to climate change 760 (Gorman, 2012), rainfall variability over the last 10-100 ky might be well represented in ECRN rates but not in kstr values which potentially integrate over longer timespans which are most likely characterized by important variations in hydrology. Moreover, we use hydrological data integrating over "only" 35 years to constrain the distribution of river discharge: these data are unlikely to fully capture rainfall variability over the integration timespan of ECRN measurements. Different integration timespans of river profile response, ECRN rates and hydrological data can be expected to affect model performance.

765 While our dataset does not enable us to fully capture rainfall variability, a distinction can be made between temporal and spatial variations. Contrary to temporal variations controlling frequency and magnitude of discharge events, the spatial gradient in orographic precipitation is(Campforts et al., 2017; Goren et al., 2014; Snyder et al., 2003; Whipple, 2001; Wobus et al., 2006). During both of these time scales it is unlikely that the temporal rainfall distribution that we inferred from 35 years of data remained stationary. Thus, there is little reason to believe that our data fully capture rainfall variability over the response times of river profiles and hillslopes. Contrary to temporal variations, spatial patterns in orographic precipitation are 770 characteristic to the formation of a mountain range at geological timescales (Garcia-Castellanos and Jiménez-Munt, 2015). In

750

the case of the Southern Ecuadorian Andes, orographic precipitation results from moist air advection via the South American Low-Level flow generates pronounced patterns of orographic precipitation (Campetella and Vera, 2002). The air is lifted as it passes over the eastern flanks of the Andes, resulting in moist convection fuelled by adiabatic decompression. Onset of 775 Andean uplift in Ecuador has been reported to be asynchronous from south to north with the onset of the most recent uplift phase dated back to These patterns likely persisted since at least the most recent uplift phase of Andean uplift in the Late Miocene (Spikings et al., 2010; Spikings and Crowhurst, 2004). Climate changes over the Miocene Pliocene probably altered absolute amounts of precipitation in the Ecuadorian Andes (Goddard and Carrapa, 2018) challenging the use of present dayrunoff and discharge distribution to predict long term river incision. However, the orographically induced gradients in 780 precipitation must have been present for timescales exceeding those represented by both km and ECRN. This partly explains why accounting for spatial variations in precipitation does improve the performance of a stochastic threshold SPM contrary to the use of catchment specific discharge distributions representing temporal discharge variability. Present-day rainfall and runoff gradients (Figure 6) are thus deemed to be representative for times exceeding response times of longitudinal river profiles and integration times of CRN-derived denudation rates, and warrant the use of contemporaneous runoff data to 785 represent spatial patterns of discharge (section 3.1). Ultimately, performance of the different stream power models underscores this interpretation. While accounting for spatial patterns in runoff improves the performance of a Stochastic-Threshold SPM (Table 4 and section 4.2.2), incorporating proxies of temporal discharge variability leads to no improvement of model performance (the role of k in section 4.2.2).

Downscaling the WRR2 WaterGAP reanalysis dataset by amalgamating regional rain gauge data, allowed to obtain a runoff dataset at a resolution suitable for use in our study. However, to further improve the accuracy of hydrological data, the use of more advanced methods might be considered. A possible approach is the application of regional climate models (e.g. Thiery et al., 2015) in regions with pronounced topographic and climatological gradients. Regional climate models have been shown to simulate rainfall variability more realistically than global re analysis datasets in mountainous areas (Thiery et al., 2015) and have been successfully used to explain geomorphic response in such areas (Jacobs et al., 2016).

795

800

805

**7.2.5.2. Environmental control on long term river incision rates**

**7.2.1.5.2.1. Geology**

Incorporating rock strength variability when simulating river incision improves model efficiency for all evaluated SPMs (Table 4 and Table 5). Our results corroborate earlier findings that established functional dependencies between river incision and rock physical properties to successfully determine river incision ratesIn all our simulations, model efficiency improves when incorporating rock strength variability (Table 4), which is consistent with earlier studies (Lavé and Avouac, 2001; Stock and Montgomery, 1999). In this study, rock strength is represented bythe absence of generally accepted metrics of erodibility, we employ an empirically derived lithological erodibility index ( $L_E$ , Eq. (15))?) based on the-age and the-lithological composition of stratigraphic units. Because ofOwing to its simplicity, our empirical approach holds potential to this or a similar index can potentially be applied at continental to global scales where detailed information on rock physical properties are not alwaysusually lacking the detail available. However, at smaller spatial scales, studies evaluating the role of rock strength heterogeneity on specific river incision processes such as fluvial abrasion will benefit from a more mechanistic approach to quantify rock strength (Attal and Lavé, 2009; Nibourel et al., 2015). MoreoverNotwithstanding, river incision efficacy might also dependdepends on other rock properties such as the density of bedrock fractures, joints and other discontinuities (Whipple

- 810 et al., 2000). Fracture density has in turn bebeen linked to spatial patterns of seismic activity (Molnar et al., 2007). Given the limited variability of seismic activity within the Paute basin (Petersen et al., 2018), seismicity was not considered in our statistical regional analysis but should be considered when applying our approach to other regions prone to more(Petersen et al., 2018 Figure S2), seismicity was not considered in our statistical regional analysis but could be considered when applying our approach to other regions characterized by more spatial seismic variability.
- We show that consideringIncorporating spatial patterns of rock strength variability not only reduces the scatter surrounding the modelled river incision versus ECRN-derived erosiondenudation rates, but also controls the degree of the nonlinearity between river steepness (*ksn*) and erosiondenudation rates, expressed by the slope exponent *n* coefficient in the A-SPM (Figure 12). When not considering(Figure 7). Omitting rock strength variability, the results in a *ksn*-ECRN relationship relation that is close to being a-linear one forin the Paute eatchmentscatchment (
[revised manuscript text omitted]
 the combination of spatially varying runoff explain part of the variability of and incision thresholds explains the observed-erosion rates. The, non-linear relationship. We do not detect, however, an impact on river incision of temporal variations in discharge, controlling the magnitude and frequency of fluvial discharge, could not be identified within the studied catchments, distributions on river incision. We attribute this partly to the limited CRN dataset but mainly to the lack to the integration time of rainfallCRN data which integrate over sufficiently longand response times of river longitudinal profiles which extend beyond timescales at which discharge distributions can be assumed to be recorded in

915

920

the CRN derived erosion rates.stationary.

890

895

900

Our study shows the potential of a stochastic threshold stream power model as a tool to explainST-SPM to infer regional and, potentially, continental to global differences in rainfall variability. However, the latter will only be successful after elucidating the confounding role of we emphasize that its application needs to account for other environmental variables such as rock strength-on river incision rates. Simplifications involved with the use of any Stream Power-. Simplified process representation of stream power-based incision model such as the models (e.g., lack of sediment-bedrock interactions-or dynamic channel width adjustments) potentially might explain part of the remaining scatter surroundingbetween predicted versusand measured erosiondenudation rates. However, residual analysis showed that most of the remaining scatter occurs in small transient catchments (up to 10 km2). To2) where sporadic mass wasting processes on hillslopes likely obscure the relation between our measurements and predictions. Elucidating this relation further our understanding of landscape evolution over different spatial scales in such transient regions, we propose the development of process-basedis potentially fostered by dynamic numerical landscape evolutions models which explicitly simulatingsimulate the coupling between transient river adjustment and stochastic-hillslope response.

**930**

925

**Data availability.**

All data used in this paper is freely available from referenced agencies. Hydrological data is available from earth2observe.eu and http://www.serviciometeorologico.gob.ec/biblioteca/. Topographic data is available from NASA (NASA JPL, 2013). Lithological data is provided in the supplementary information. Calculations were done in MATLAB using the TopoToolbox Software (Schwanghart and Scherler, 2014).

**935**

.

**Author contribution.**

InBC conceived the project in collaboration with all the authors, BC designed the project, carried outVV, MVM and GG.
 BC performed the statistical analysis and took the numerical calculations and wrotelead in writing the manuscriptpaper, All authors contributed to editingshaping the manuscript. research and analyses, as well as writing the paper.

**Competing interests.**

The authors declare that they have no conflict of interest.

**945 Financial support.**

B. Campforts received a postdoctoral grant from the Research Foundation Flanders (FWO)). G. Tenorio received a PhD grant from ARES – the Académie de recherche et d'enseignement supérieur de la Belgique - as part of the PRD research cooperation project: Strengthening the scientific and technological capacities to implement spatially integrated land and water management schemes adapted to local socio-economic, cultural and physical settings, and a grant from the Secretaría de

950 Educación Superior de Ciencia, Tecnología e Innovación de la República del Ecuador.

Field Code Changed

| Formatted: English (United Kingdom) |
|-------------------------------------|
| Formatted: English (United Kingdom) |

**9.7.References**

| i i |                                                                                                                                                                                                                                                                                                                                                |                                     |
|-----|------------------------------------------------------------------------------------------------------------------------------------------------------------------------------------------------------------------------------------------------------------------------------------------------------------------------------------------------|-------------------------------------|
|     | Aalto, R., Dunne, T. and Guyot, J. L.: Geomorphic Controls on Andean Denudation Rates, J. Geol., 114(1), 85–99,
doi:10.1086/498101, 2006.                                                                                                                                                                                                   | Formatted: English (United Kingdom) |
| 955 | Abbühl, L. M., Norton, K. P., Jansen, J. D., Schlunegger, F., Aldahan, A. and Possnert, G.: Erosion rates and mechanisms of knickzone retreat inferred from 10Be measured across strong climate gradients on the northern and central Andes Western Escarpment, Earth Surf. Process. Landforms, 36(11), 1464–1473, doi:10.1002/esp.2164, 2011. |                                     |
| 960 | Alcamo, J., Döll, P., Henrichs, T., Kaspar, F., Lehner, B., Rösch, T. and Siebert, S.: Development and testing of the WaterGAP 2 global model of water use and availability, Hydrol. Sci. J., 48(3), 317–337, doi:10.1623/hysj.48.3.317.45290, 2003.                                                                                           |                                     |
|     | Anderson, R. S.: Evolution of the Santa Cruz Mountains, California, through tectonic growth and geomorphic decay, J. Geophys. Res., 99(B10), 20161–20179, doi:10.1029/94JB00713, 1994.                                                                                                                                                         |                                     |
|     | Armitage, J. J., Whittaker, A. C., Zakari, M. and Campforts, B.: Numerical modelling of landscape and sediment flux response to precipitation rate change, Earth Surf. Dyn., 6(1), 77–99, doi:10.5194/esurf-6-77-2018, 2018.                                                                                                                   |                                     |
| 965 | Attal, M. and Lavé, J.: Pebble abrasion during fluvial transport: Experimental results and implications for the evolution of the sediment load along rivers, J. Geophys. Res., 114(F4), F04023, doi:10.1029/2009JF001328, 2009.                                                                                                                |                                     |
|     | Basabe R, P.: Geologia Y Geotecnia, in Prevencion de Desastres Naturales en la Cuenca del Paute. Informe Final: Projecto
PreCuPa. Swiss Disaster Relief Unit (SDR/CSS), pp. 1–153, Cuenca, Ecuador., 1998.                                                                                                                                  | Formatted: English (United Kingdom) |
| 970 | Beek, H. E., Van Dijk, A. I. J. M., Levizzani, V., Schellekens, J., Miralles, D. G., Martens, B. and De Roo, A.: MSWEP: 3-
hourly 0.25° global gridded precipitation (1979-2015) by merging gauge, satellite, and reanalysis data, Hydrol. Earth Syst.
Sei., 21(1), 589-615, doi:10.5194/hess-21-589-2017, 2017.                         |                                     |
|     | Bendix, J., Rollenbeck, R., Göttlicher, D. and Cermak, J.: Cloud occurrence and cloud properties in Ecuador, Clim. Res., 30, 133–147, doi:10.3354/cr030133, 2006.                                                                                                                                                                              | Formatted: English (United Kingdom) |
| 975 | Bennett, G. L., Miller, S. R., Roering, J. J. and Schmidt, D. A.: Landslides, threshold slopes, and the survival of relict terrain in the wake of the Mendocino Triple Junction, Geology, 44(5), 363–366, doi:10.1130/G37530.1, 2016.                                                                                                          |                                     |
|     | Bernhardt, A., Schwanghart, W., Hebbeln, D., Stuut, JB. W. and Strecker, M. R.: Immediate propagation of deglacial environmental change to deep-marine turbidite systems along the Chile convergent margin, Earth Planet. Sci. Lett., 473, 190–204, doi:10.1016/j.epsl.2017.05.017, 2017.                                                      |                                     |
| 980 | Binnie, S. A., Phillips, W. M., Summerfield, M. A. and Fifield, L. K.: Tectonic uplift, threshold hillslopes, and denudation rates in a developing mountain range, Geology, doi:10.1130/G23641A.1, 2007.                                                                                                                                       |                                     |
|     | Bishop, P., Hoey, T. B., Jansen, J. D. and Artza, I. L.: Knickpoint recession rate and catchment area: the case of uplifted rivers in Eastern Scotland, Earth Surf. Process. Landforms, 778(6), 767–778, doi:10.1002/esp.1191, 2005.                                                                                                           |                                     |
|     | von Blanckenburg, F., Belshaw, N. S. and O'Nions, R. K.: Separation of 9Be and cosmogenic 10Be from environmental materials and SIMS isotope dilution analysis, Chem. Geol., 129(1–2), 93–99, doi:10.1016/0009-2541(95)00157-3, 1996.

---

## Editor Decision (ED1)

[revised manuscript text omitted]
 both of these time scales it is unlikely that the temporal rainfall distribution that we inferred from 35 years of data remained stationary. Thus, there is little reason to believe that our data fully capture rainfall variability over the response times of river profiles and hillslopes. Contrary to temporal variations, spatial patterns in orographic precipitation are characteristic to the formation of a mountain range at geological timescales (Garcia-Castellanos and Jiménez-Munt, 2015). In the Southern Ecuadorian Andes, moist air advection via the South American Low-Level flow generates pronounced patterns of orographic precipitation (Campetella and Vera, 2002). These patterns likely persisted since at least the most recent uplift phase of Andean uplift in the Late Miocene (Spikings et al., 2010; Spikings and Crowhurst, 2004). Present-day rainfall and runoff gradients (Figure 6) are thus deemed to be representative for times exceeding response times of longitudinal river profiles and integration times of CRN-derived denudation rates, and warrant the use of contemporaneous runoff data to represent spatial patterns of discharge (section 3.1). Ultimately, performance of the different 
[revised manuscript text omitted]

---

## Author Response (AR2)

We thank the editor, Simon Mudd, for his careful revision of our work and his thoughtful and constructive comments. We included all textual suggestions and shortly comment on his other remarks in the following document.

Line 60
You try to dismiss hillslope metrics here, which I don't agree with. It is true that the hillslope metrics that do work (curvature, Hurst et al 2012) or rock exposure (DiBiase et al 2012) require high resolution topographic data that is not widely available.
We agree and changed the sentence to: In such a configuration, hillslope gradients are no longer an indication of denudation rates, and hillslope metrics (Hurst et al., 2012) often require high resolution topographic data that are not widely available.

Line 62
Hillslope *gradients*. There are plenty of hillslope metrics that are sensitive to erosion rates (e.g., ridgetop curvature or fraction of rock exposure)
Agreed, adjusted the sentence accordingly

Line 84
You mean observational records in mountain regions? Clarify. Many lowland rivers have long and complete hydrological records.
Correct, it is now clarified by specifying "in mountain regions"

Line129
I would be careful here: in the hydrology literature and some of the early geomorphology literature the concavity of the profile is measured (for example, see the Chen et al Nature paper that came out recently). This is the concavity *index*
Agreed, adjusted to concavity *index*

Line 189 Is it still rising? If so does the model try to account for this?
We added a sentence: Uplift patterns are assumed to be reflected in the river steepness and not explicitly simulated in this paper.

Line 255
Uncertainties associated to the WaterGAP3 data originate from hydrological model assumptions and spatially distributed input data (Beck et al., 2017). We revisit the impact of uncertainties on the climatological data on our model runs in the discussion of this paper.

Line 316
Add reference here.
Done

I don't understand this. Please clarify. Why would the river incision rate inferred by a nuclide concentration be affected by runoff?
Fair point, this sentence was confusing, and has been rewritten. For the CRN data, one assumes that the catchments are in isotopic steady state – that the input of CRN by in-situ production equals the export of CRN by fluvial processes, and radioactive decay. For the river incision models, one uses one value of precipitation and runoff data per catchment – and assumes that the pattern is rather uniform over the catchment.

Line 374:
For completeness, I would add the equation for this since you have it for ME and NS
Agreed, done

Line 413: Interesting. The gasparini and brandon paper shows that thresholding effects in the SPM can be approximated by n>1. It is a theoretical result that anticipates your result. I think this can be more clearly explained around line 480 (see more comments below).

Here I suggest referring to the empirical studies that suggest n>1 (the papers you cite in a similar discussion on line 576).

Good suggestion to cite the empirical work at this stage.

The paper of Gasparini and Brandon 2011 mainly focuses on the influence of sediment fluxes on bedrock river incision. They do not explicitly simulate the role of thresholds and we feel that the theoretical framework laid out in the papers we cite in the introduction paragraphs are providing the right background for evaluating and simulating the role of thresholds (e.g. Deal et al., 2018; Lague et al., 2005; Tucker and Bras, 2000)

Line 425: There isn't a figure showing a relationship between E_CRN and k_sn. You are trying to argue that lithological heterogeneity is masking a more interesting pattern, but you don't show the data that would allow the reader to make this interpretation. In addition, I don't think this paragraph is really true to your results. What I see is this: as long as you force n = 1, the fits result in NS = 0.5, R^2 = 0.5. The fit is very slightly better if you use spatially heterogeneous lithology, rather than basin averages. But the best way to increase the fit is to let n vary (and when you do that, n>1.

We believe there is a small misunderstanding here:
1. In all model runs, lithological variability is either simulated using a fixed constant value or the average values of the individual sub-catchments ($\overline{L_E}$ values in Table 2). To clarify this, we added the following lines in the methodology, below Eq. 11:
   *"Note that, at any point in the paper, lithological heterogeneity within the Paute catchment is represented using the average values of $L_E$ for the individual sub-catchments indicated with $\overline{L_E}$ and listed in Table 2. If lithological heterogeneity is not considered, $\overline{L_E}$ is fixed to a value of 1. "*
   If lithological heterogeneity is not considered, none of the models with n>1 (scenario 1) or *n*=1 (scenario 3) can successfully predict the E_CRN derived erosion rates. Only when lithological heterogeneity is considered, the goodness-of-fit of the models increases. The best fit is then obtained when *n* is larger than 1, i.e. with *n* =1.64 (scenario 2). This is indeed a key message of the paper, so we tried to clarify this by:
       a) *Adjusting the overview plot (*Figure 9): By adding the grey bars also for ME, it should be clear that none of the models performs well if lithological heterogeneity is not considered
       b) We added the *k_sn* versus ECRN plot and the *k_sn* versus ECRN/LE plot to show the actual relationships between E_CRN and *k_sn* (new subplots a and b in Figure 7)
       c) We adjusted this piece of text by removing the reference to the model scenario and now refer directly to the subplots showing the *k_sn* -E_CRN relationships

Line 445:
This is predicted by the gasparini and brandon paper
We will be careful with referring to the Gasparini and Brandon 2011 paper: they do not take runoff explicitly into account in their model simulations (they consider models that include channel gradient, sediment flux, and drainage area). We therefore prefer not citing this paper at this stage to avoid confusion.

Line 498
So basically, you can do a ton of data gathering and make a very fancy runoff/thresholding stream power model, and it doesn't do any better than adjusting the n exponent (as predicted by Gasparini and Brandon). Is that right?
We only partially agree here:
- We have clearly shown that lithological variability is key to consider. Regardless the value of *n*, if lithological heterogeneity is not considered, *n*>1 does not help to increase the goodness-of-fit of the models. See also previous comments, adjustments to figure 7 and figure 9
- Including thresholds and spatially variable runoff has a similar effect than a value of *n*>1. The work of Lague 2014 summarizes the theory supporting this. So, in terms of fitting measured erosion rates, we agree with the editor. However, explicitly incorporating and calibrating the role of thresholds and runoff variability, helps to understand and hence predict the role of thresholds and climate variability in landscape evolution.

Line 555
So you argue that the lithological variability is important. But it looks like a basin averaged approach is not worse than using the geologic map for A-SPM. Would it be useful to conduct a sort of straw man experiment: run the model with a *single* erodibility and optimize for this parameter. My intuition is that this model fit would be terrible. But it

would go some way to demonstrate that yes, you have to account for lithology. But a basin averaged approach is probably good enough. Or have I misinterpreted your results?

We believe there is a small confusion here. All calculations are performed using a basin average approach. With lithological heterogeneity, we actually refer to the use of catchment average values as reported in Table 2. We added some words in the methodology and adjusted figures 7 and 9 to clarify this. See also reply to previous suggestions of the editor.

Line 607 Was there also not a paper by Ferrier et al in Hawaii that said something like this?

Indeed, we already cited this work but not at this place. Thanks for the suggestion.

[revised manuscript text omitted]

---

## Author Response (AR3)

We thank the associate editor, Simon Mudd for his insightful comments which significantly helped improving the quality of our paper. In the following we shortly document the final changes we made to the manuscript in response to the suggestions of the AE.

Thanks for the revisions. Thinks are much clearer now. I have a few minor technical issues remaining before this gets sent to the copyeditors.

I was indeed confused: I thought you used the actual lithology map when you simulated the lithologically heterogeneous sites. I now understand what was done (I think).
Did you try to run the models using the actual lithology values (that is, different K values for each pixel) from the map?
Indeed, in all the runs, we use catchment average values. We did not do any runs with actual lithological values per pixel. In the methodology section (3.2), this is now explicitly acknowledged in the sentences:
*Note that, at any point in the paper, lithological heterogeneity within the Paute catchment is represented using the average values of $L_E$ for the individual sub-catchments indicated with $\overline{L_E}$ and listed in Table 2. If lithological heterogeneity is not considered, $\overline{L_E}$ is fixed to a value of 1.*

The table is much clearer with the footnotes, but I still find it slightly confusing to have "L_E overbar fixed" in the table. I think it would be clearer to say "L_E = 1" here. Is there a reason that wasn't done?
Good suggestion to actually write L_E overbar = 1 in the table. We adjusted the table accordingly.

Why is it called L_E in many places but in table 3 it is called L_L?
The L_L value in table 3 refers to the L_L in Eq. 7. I will add a reference to Eq. 7 in the table caption to clarify this.

Table 2: commas are used instead of full stops (i.e., the table use the continental European convention instead of English language convention). This needs to be changed
Thanks for pointing us to the use of commas, we changed them in points

[revised manuscript text omitted]